# Reward regulation in plant–frugivore networks requires only weak cues

Jörg Albrecht[1], Jonas Hagge [2], Dana G. Schabo[3], H. Martin Schaefer[4] & Nina Farwig[3]

Theory assumes that fair trade among mutualists requires highly reliable communication. In plant–animal mutualisms the reliability of cues that indicate reward quality is often low. Therefore, it is controversial whether communication allows animal mutualists to regulate their reward intake. Here we show that even loose relationships between fruit brightness and nutritional rewards ($r^2 = 0.11$–$0.35$) allow birds to regulate their nutrient intake across distinct European plant–frugivore networks. Resident, over-wintering generalist frugivores that interact with diverse plant species select bright, lipid-rich fruits, whereas migratory birds select dark, sugar- and antioxidant-rich fruits. Both nutritional strategies are consistent with previous physiological experiments suggesting that over-wintering generalists aim to maximize their energy intake, whereas migrants aim to enhance the build-up of body fat, their immune response and oxidative status during migration. Our results suggest that animal mutualists require only weak cues to regulate their reward intake according to specific nutritional strategies.

[1] Senckenberg Biodiversity and Climate Research Centre (BiK-F), Senckenberganlage 25, 60325 Frankfurt am Main, Germany. [2] Department of Zoology, Entomology Research Group, Technical University of Munich, Hans-Carl-von-Carlowitz-Platz 2, 85354 Freising, Germany. [3] Conservation Ecology, Faculty of Biology, Philipps-University Marburg, Karl-von-Frisch Str. 8, 35043 Marburg, Germany. [4] Fundación Jocotoco, Lizardo García E9-104 y Andrés Xaura, P.O. Box 17-16-337, Quito, Ecuador. Correspondence and requests for materials should be addressed to J.A. (email: joerg.albrecht@senckenberg.de)

Mutualistic networks are characterized by the repeated exchange of resources and services between species[1]. Communication is thought to be a key component of trade within these networks because the involved species often possess traits that are adapted to stimulate the sensory system of their mutualistic partners[2]. In pollination and seed dispersal networks, for instance, where plants provide rewards (e.g., nectar and fruit pulp) in exchange for dispersal of pollen and seeds by animals, flower and fruit displays usually produce visual and olfactory stimuli that are adapted to the sensory system of animal mutualists[3]. A fundamental question is whether these stimuli mainly serve to attract animal mutualists via increased conspicuousness[3], or whether they also provide information about reward quality that animal mutualists use to regulate their reward intake by selecting those plants that best match their nutritional demands[4,5].

Signalling theory assumes that mutualists require highly reliable cues (i.e., a strong correlation between cue and reward quality) to regulate their reward intake[6,7]. Yet, in most mutualisms cue–reward relationships are weak[5,8], suggesting that the reliability of cues may be too low to inform reward regulation. More recent theoretical work[9] highlights, however, that animal mutualists may be able to regulate their reward intake, despite low reliability, by verifying the accuracy of cues during repeated interactions and by abandoning plants whose rewards do not match expectations from cues[5,10,11]. This mechanism has been largely overlooked in signalling theory because most theoretic models assume that individuals interact with each other only once or hold no memory of previous interactions[9,12]. Despite the high prevalence and diversity of communicative traits in mutualisms[2], it is therefore unresolved to what extent communication contributes to reward regulation in mutualistic networks[7].

Seed dispersal mutualisms between fleshy-fruited plants and frugivorous animals represent an exceptional opportunity to address this question[4]. The macronutrient composition of fleshy fruits (i.e., the lipid, sugar and protein content of fruit pulp) is often unbalanced and does not necessarily match the nutritional requirements of frugivores[13,14]. Therefore, frugivores need to actively balance and regulate their nutrient intake by consuming fruits of different plant species[15]. Nutritional strategies of frugivores, in turn, may vary depending on specific requirements associated with their biology and life history. For example, some frugivore species within plant–frugivore networks are extreme generalists that have a strong impact on network dynamics and seed dispersal processes[1,16]. These generalists do not only show morphological, behavioural, and physiological adaptations that allow them to consume fruits of a diverse set of plant species[17–19], but they also strongly depend on fruit resources[20,21]. According to optimal foraging theory, these frugivores should select high-caloric lipid-rich fruits to maximize their net energy gain when relying mainly on fruit resources[14,22].

In addition, many frugivorous birds are migrants[23,24] that on their journey between their breeding grounds in temperate and their wintering grounds in tropical and subtropical latitudes rely almost exclusively on energy stored as body fat[25]. Successful migration thus depends on the amount of fat accumulated prior to migration and on the rate of fat deposition during stopover[25,26]. Field studies at migratory stopover sites indicate that frugivory allows for a more efficient and more extensive gain of body fat than insectivory[27]. This has mainly been attributed to the fact that the lower protein to calorie ratio of fruits, compared to insects, may facilitate fat deposition[14,28]. More detailed recent experiments have shown that, when dietary protein content is low, especially diets with high-sugar content enhance fat deposition in migratory birds (via hepatic de novo lipogenesis), whereas isoenergetic diets with high-lipid content rather stimulate the direct utilization of dietary fat[28]. Therefore, birds on high-sugar, low-protein diets gain more body fat than birds on high-lipid, low-protein diets[28]. In light of these experiments, migratory birds should select sugar-rich fruits to enhance the accumulation of body fat[28], whereas resident over-wintering birds should select lipid-rich fruits to maximize their net energy gain. Moreover, migrating birds are exposed to high oxidative stress associated with fat oxidation during flight[29], and their innate immune function is compromised by physiological and energetic trade-offs[30]. As the intake of certain fruit pigments such as anthocyanin reduces oxidative stress and stimulates the immune response of birds[31], migratory birds might select anthocyanin-rich fruits to enhance their antioxidant capacity and immune response[29–34]. However, even though fruit colours are adapted to the visual system of frugivores[3], and visual and nutritional fruit traits are commonly correlated[35,36], it is still unknown whether frugivores use fruit colours as cues to regulate their reward intake according to the above-described nutritional strategies.

Here we address this question by applying concepts from network theory[1], signalling theory[4] and nutritional ecology[15] to ten seasonally resolved plant–frugivore networks from three distinct European localities including 159,588 interactions between 44 plant and 43 bird species (Fig. 1; Methods; Supplementary Table 1). First, we assess how reliable colour-reward relationships of fleshy fruit displays are. Second, we test whether fruit choice of frugivorous birds is mediated by fruit colour and whether the frugivores' mean intake of particular nutrients is related to the diversity of their interaction partners (partner diversity), to their relative contribution to fruit removal in the plant–frugivore networks (interaction strength) and to their migratory behaviour (the latitudinal migratory distance of a frugivore; see Methods for details). To answer these questions we use an integrative community-wide modelling approach in a Bayesian hierarchical framework. Thereby, our analysis takes advantage of the fact that generalist and specialist frugivores, as well as migratory and resident frugivores co-occur within the seasonal networks and have access to the same fruiting plant species. This setting represents a natural experiment that allows for a comparison of nutritional strategies between these groups, because interspecific differences in fruit selection are unlikely to be confounded by spatiotemporal constraints in resource availability. We find weak associations between fruit brightness and nutritional rewards. Moreover, we discover that, consistent with physiological experiments, the reward intake of frugivorous birds is related to their partner diversity and migratory behaviour and is primarily mediated by fruit brightness. These results indicate that even weak cue–reward relationships may allow animal mutualists to regulate their reward intake according to specific nutritional strategies.

## Results

**Colour–reward relationships.** To assess how reliable colour–reward relationships of fleshy fruit displays are, we first quantified fruit colours according to the visual perception of birds in tetrahedral colour space[37,38] (Fig. 2a, b; see Methods). The avian colour space within the tetrahedron is characterized by three Cartesian coordinates ($x$, $y$, $z$) that are related to particular chromatic colour components. An achromatic component ($a$) indicates whether a fruit is perceived as being bright or dark (fruit brightness hereafter). We tested whether fruit colour, as perceived by birds, is related to the nutrient content of fruits using a Bayesian hierarchical model with a stochastic variable selection procedure that included a phylogenetic random factor to account for the phylogenetic relatedness of the plant species (Fig. 2c; see

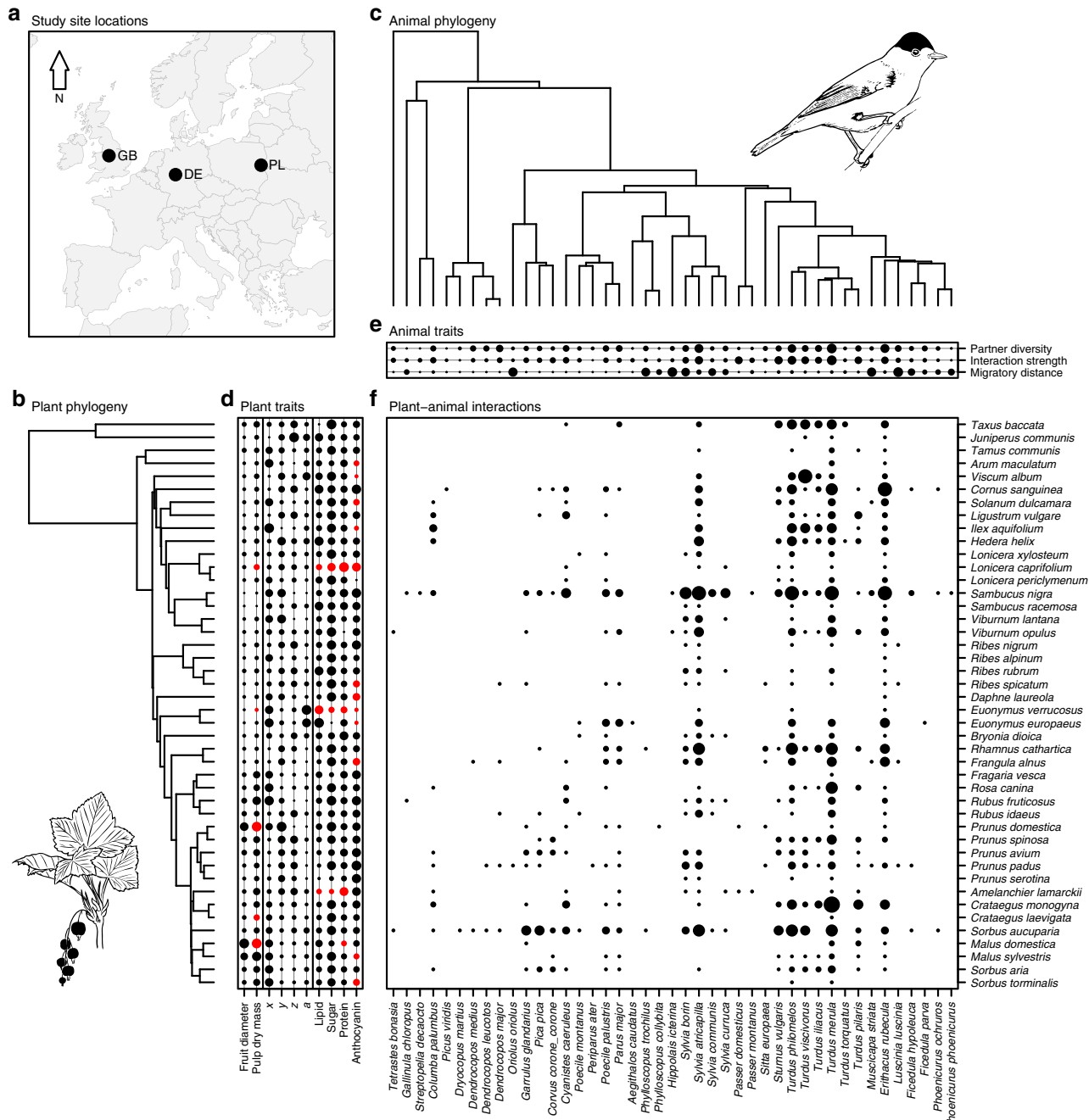

**Fig. 1** Summary of the data from the three European plant–frugivore associations that have been used in the present study. **a** The locations of the three study sites, **b**, **c** the plant and animal phylogenies, **d**, **e** plant and animal traits, and **f** the pooled metaweb of plant–frugivore interactions (see Methods for details). The dots in **a** represent the locations of the plant–frugivore associations included in this study (GB: Snow and Snow[23]; DE: Stiebel and Bairlein[58]; and PL: Albrecht et al.[59]). In **d**, **e** traits are scaled to the interval [0, 1], with larger dots indicating a larger relative trait value. In **d** (x, y, z) and (a) characterize the fruit colours of the plant species according to the visual perception of birds in tetrahedral colour space[37,38], whereby (x, y, z) represent the chromatic components of fruit colouration and (a) represents fruit brightness (i.e., the achromatic colour component). Red dots in **d** indicate plant traits that have been inferred using Bayesian data augmentation (see Methods). In **e** the mean trait values of frugivores are shown for simplicity, although for a given frugivore species a trait could vary across the three study sites and across seasons (see Methods). In **f** the size of the dots depicting plant–frugivore interactions indicates the proportional occurrence of a given interaction across sites and seasons (with larger dots representing interactions with higher spatiotemporal constancy). Note that even though we present the pooled metaweb as a summary in **f**, we explicitly accounted for the fact that individual networks were sampled in different localities and during different seasons in the analysis. The map in **a** was created in R[75] using the package *rworldxtra*[79]. Made with Natural Earth. The drawings of black currant (*Ribes nigrum*) in **b** and Eurasian blackcap (*Sylvia atricapilla*) in **c** were created by J. Albrecht

Methods). To quantify the reliability of the colour–reward relationships, we used marginal $r^2$-values ($r_m^2$)[39] that measure the variation in nutrient content that is explained by fruit colour while accounting for the phylogenetic relatedness of plants.

Across the three localities, the nutritional rewards of fruits were consistently related to fruit brightness, but not to the chromatic colour components of the avian colour space (Table 1; Fig. 2c). Fruit brightness indicated with low to moderate reliability the

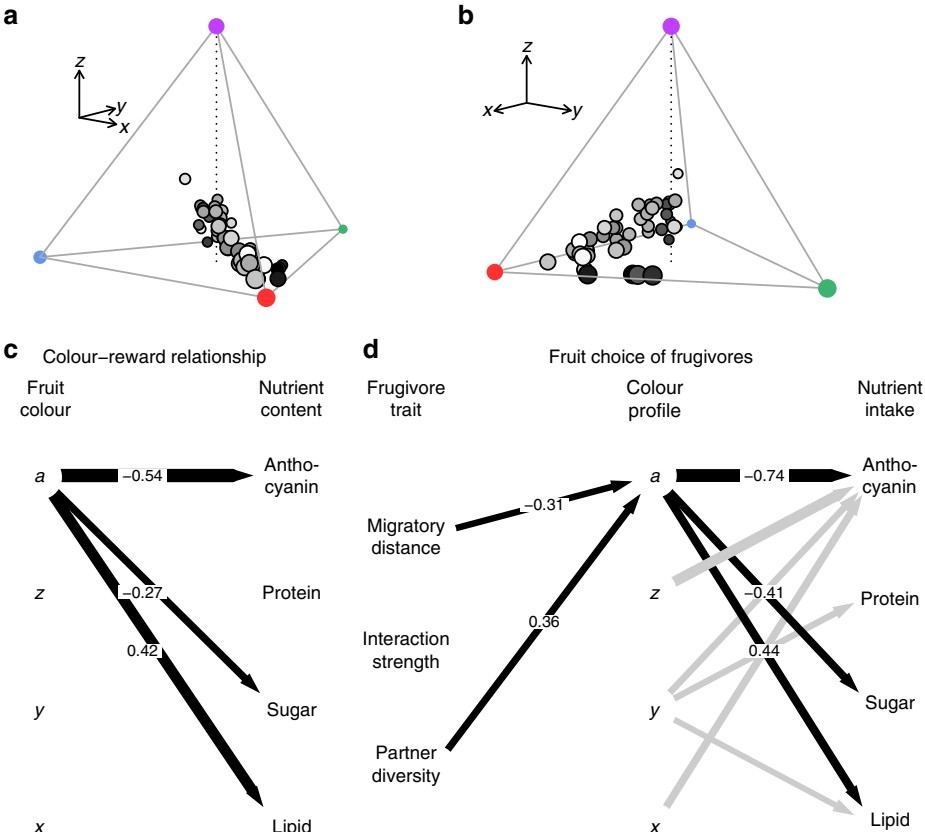

**Fig. 2** Colour–reward relationships mediate fruit choice by frugivorous birds. **a**, **b** Fruit colours of the 44 plant species in the plant–frugivore networks according to the visual perception of birds in tetrahedral colour space[37,38] from two different perspectives (see Methods). The avian colour space within the tetrahedron is characterized by three Cartesian coordinates ($x$, $y$, $z$) that are related to particular chromatic colour components. The $x$-coordinates range from blue (negative scores) to red (positive scores), $y$-coordinates range from purple (negative scores) to green (positive scores), whereas $z$-coordinates indicate UV reflectance (positive scores). The achromatic component (fruit brightness, $a$) indicates whether a fruit is perceived as being bright (high values) or dark (low values). In **a**, **b** fruit brightness is indicated by the grey colour of the circles, ranging from white (bright fruits) to black (dark fruits). **c** Most parsimonious Bayesian hierarchical model showing the relationships between chromatic colour components of fruit colours ($x$, $y$, $z$), fruit brightness ($a$) and the nutrient concentrations in fruit pulp (lipid, sugar, protein, and anthocyanin). The sample size in **c** was $n_{species} = 44$ plant species. **d** Most parsimonious Bayesian hierarchical structural equation model of fruit choice by frugivores showing the relationships of the birds' partner diversity and interaction strength in the networks and their migratory behaviour with the colour profile of consumed fruits and the mean intake of particular nutrients. The sample size in **d** was $n_{obs} = 165$ observations across $n_{species} = 43$ bird species, $n_{site} = 3$ study sites and $n_{time} = 4$ seasons. In **c**, **d** relationships that received positive support during the variable selection (BF > 2) are highlighted in black. See caption of Table 1 for interpretation of BF-values. The numbers on the arrows indicate the effect size and direction of the relationships. In **d** relationships between colour profiles of consumed fruits and the nutrient intake of birds that were supported by the variable selection but were not related to frugivore traits are drawn in grey

lipid ($r_m^2 = 0.23$, $2\log_e$(Bayes Factor) = 6.0, strong support, BF hereafter), sugar ($r_m^2 = 0.11$, BF = 2.5, positive support) and anthocyanin content of fruits ($r_m^2 = 0.35$, BF = 6.8, strong support). Dark fruits contained higher sugar and anthocyanin concentrations than bright fruits, whereas bright fruits contained higher lipid concentrations than dark fruits (Fig. 2c). There was no relationship between fruit brightness and the protein content of fruits ($r_m^2 = 0.03$, BF = −1.5, no support).

**Community-wide fruit choice and nutrient regulation**. We tested whether the fruit choice of birds is mediated by fruit colour and whether the birds' mean intake of particular nutrients is related to their partner diversity and interaction strength in the plant–frugivore networks and to their migratory behaviour. To do so, we used a Bayesian hierarchical structural equation model with a stochastic variable selection procedure (Fig. 2d; see Methods). The model contained a phylogenetic random factor to account for the phylogenetic relatedness of the bird species, as

well as random factors for species, site and season to account for the hierarchical structure of the data.

The model indicated that the inherent relationship between fruit brightness and nutritional rewards observed across the sampled plant species was also present in the diets of birds, because the mean intake of lipid, sugar and anthocyanin was correlated with the mean brightness of consumed fruits (lipid: $r_m^2 = 0.14$, BF > 15, decisive support; sugar: $r_m^2 = 0.09$, BF = 14, decisive support; anthocyanin: $r_m^2 = 0.48$, BF > 15, decisive support; Table 2). In contrast, the mean intake of protein was not related to the mean brightness profile of consumed fruits ($r_m^2 = 0.04$, BF = −1.9, no support; Table 2).

Importantly, the analysis also revealed that the partner diversity and migratory behaviour of birds were exclusively related to the mean brightness of consumed fruits but not to the chromatic colour components of the avian colour space (Fig. 2d). This is in line with the analyses of the colour–reward relationships across the 44 plant species (Table 1). Across the three localities, frugivores that integrated fruits of a high diversity of

**Table 1 Summary of Bayesian hierarchical model**

| Response ~ predictor variable | Estimate (95% CI) | P | BF |
|---|---|---|---|
| Lipid content ~ ($r_m^2 = 0.23$, $r_c^2 = 0.51$) | | | |
| x | 0.02 (−0.2, 0.3) | 0.31 | −1.6 |
| y | 0.01 (−0.1, 0.2) | 0.29 | −1.8 |
| z | 0.004 (−0.2, 0.2) | 0.27 | −2.0 |
| a | 0.4 (0, 0.7) | 0.95 | 6.0* |
| Sugar content ~ ($r_m^2 = 0.11$, $r_c^2 = 0.23$) | | | |
| x | −0.009 (−0.3, 0.2) | 0.33 | −1.4 |
| y | 0 (−0.2, 0.2) | 0.29 | −1.8 |
| z | 0 (−0.3, 0.3) | 0.31 | −1.6 |
| a | −0.3 (−0.7, 0) | 0.77 | 2.5* |
| Protein content ~ ($r_m^2 = 0.029$, $r_c^2 = 0.16$) | | | |
| x | 0 (−0.3, 0.3) | 0.34 | −1.4 |
| y | 0.04 (−0.1, 0.4) | 0.37 | −1.1 |
| z | 0.03 (−0.2, 0.4) | 0.36 | −1.2 |
| a | 0 (−0.3, 0.3) | 0.32 | −1.5 |
| Anthocyanin content ~ ($r_m^2 = 0.35$, $r_c^2 = 0.42$) | | | |
| x | −0.05 (−0.5, 0.3) | 0.44 | −0.44 |
| y | 0.2 (−0.01, 0.6) | 0.68 | 1.5 |
| z | 0.2 (−0.07, 0.6) | 0.57 | 0.60 |
| a | −0.5 (−0.9, 0) | 0.97 | 6.8* |

The model tested the relationships between the chromatic colour components (x, y, z) and the brightness of fruits (a) in the avian colour space (see Methods for details) and the lipid, sugar, protein and anthocyanin concentrations in the fruit pulp. Plant phylogeny was included as a random factor. The sample size was $n_{species} = 44$ plant species. Given are posterior means of effect estimates (with shrinkage), 95% credible intervals (CI), selection probabilities (P) and $2\log_e$(Bayes factor) (BF) as a measure of support for a given effect. BF-values < 2 indicate no support; values between 2 and 6 indicate positive support; values between 6 and 10 indicate strong support; and values > 10 indicate decisive support. Effects that were supported by the variable selection with BF > 2 are shown with an asterisk. The $r^2$ values depict the marginal ($r_m^2$) variance explained by fixed factors only as well as the conditional ($r_c^2$) variance explained by fixed and random factors combined[39]

plant–animal mutualistic networks. We find weak associations of fruit nutritional rewards with fruit brightness, but no associations with chromatic components of fruit colouration. We further discover that, consistent with expectations from previous physiological experiments, the reward intake of frugivorous birds is related to the diversity of their interaction partners in the networks and to their migratory behaviour and is primarily mediated by fruit brightness. These results suggest that frugivorous birds use the most reliable component of fruit colouration to discriminate the nutritional content of fruit pulp and that even weak cue–reward relationships may allow animal mutualists to regulate their reward intake according to specific nutritional strategies.

We found that fruit brightness was the single most important indicator of nutritional rewards across the plant species in the studied plant–frugivore networks. The covariance between fruit brightness and nutritional rewards in our study seems to be a by-product of constraints in fruit colouration due to pleiotropy and shared biochemical pathways of nutrients and pigments[35,40]. On the one hand, the accumulation of sugar in fruit pulp directly up-regulates the biosynthesis of anthocyanin, the pigments primarily imparting achromatic colouration in fruits[34,35]. On the other hand, both sugar and anthocyanin are hydrophilic, whereas lipid is hydrophobic[36]. Thus, dark fruits were rich in sugar and anthocyanin, while bright fruits were rich in lipids in our study system.

We showed that birds relied on this visual cue to regulate their reward intake according to two distinct nutritional strategies (Figs. 2d and 3). We found that resident over-wintering frugivores and generalists that consume fruits from a diverse range of plant species increased their lipid intake by consuming bright fruits. Generalist frugivores in temperate and Mediterranean ecosystems are known to switch rapidly from insectivory during the breeding season to frugivory during autumn and winter and show a strong seasonal dependency on fruit resources[18]. In line with optimal foraging theory, our results suggest that these frugivores select high-caloric lipid-rich fruits to offset the energetic costs of foraging and to meet their metabolic demands when relying mainly on fruits during autumn and winter[24,28]. In addition, our results suggest that the morphological, physiological and behavioural adaptations of generalists in plant–animal mutualisms[16–19] might also include specific nutritional strategies that enable them to rely mainly on resources that they acquire through mutualistic interactions (e.g., fruit pulp or nectar). Moreover, we found that migratory birds, unlike resident birds, increased their intake of sugar and anthocyanin by consuming dark fruits. Previous work has shown that diets that are high in sugar and low in protein enhance the accumulation of body fat in migratory birds[28]. Therefore, our results suggest that migrants may select sugar-rich fruits to minimize the high energetic costs of stop-over for refuelling[41] by increasing the amount of fat stores and the rate of fat deposition prior to migration and during stopover[25–27]. Apart from that, our findings suggest that migratory birds select anthocyanin-rich fruits to improve their oxidative status and immune response during migration[29–34]. Overall, these contrasting nutritional strategies of generalist and specialist, as well as resident and migratory frugivores contribute to explaining the considerable variation in colour preferences of avian frugivores that has been reported in previous studies[42–44]. Thereby, the link between nutrient regulation and communication identified in our study may contribute to explaining the diversity of flower and fruit displays in pollination and seed dispersal mutualisms[40]. If animal mutualists use cue–reward relationships to regulate their reward intake, selective pressures associated with diverging nutritional strategies might have contributed to a parallel diversification of plant reproductive displays and the nutrient composition of plant rewards. More generally,

plant species into their diets, as well as resident over-wintering frugivores selected bright fruits and thereby increased their intake of lipid (Fig. 2d). In contrast, specialized and migratory frugivores selected dark fruits and thereby increased their intake of sugar and anthocyanin. The interaction strength of birds in the networks was neither directly nor indirectly related to their nutrient intake (Fig. 2d).

Finally, we aimed to formally assess whether birds used the most reliable component of fruit colouration to discriminate the nutrient content of fruit pulp and to regulate their nutrient intake. To do so, we quantified for each colour component how often it was selected as a predictor of nutrient content in fruit pulp across the plant species in the first analysis (Table 1), and how often it was selected as being related to the bird's partner diversity and interaction strength in the networks and to their migratory behaviour in the second analysis (Table 2). The former gives an indication of the relative importance of a colour component for colour–reward relationships, the latter provides information about the relative importance of a colour component as mediator of fruit choice by birds (see section on Statistical analysis in the Methods). We found a strong positive relationship between both selection probabilities suggesting that birds used the most reliable component of fruit colouration to evaluate the nutrient content of fruits and regulate their nutrient intake (Fig. 3a).

### Discussion

Our study provides a community-wide assessment of the importance of cue–reward relationships for reward regulation in

**Table 2 Summary of Bayesian hierarchical structural equation model**

| Response ~ predictor variable | Estimate (95% CI) | P | BF | Response ~ predictor variable | Estimate (95% CI) | P | BF |
|---|---|---|---|---|---|---|---|
| $x$ ~ ($r_m^2 = 0.0025$, $r_c^2 = 0.75$) | | | | Lipid intake ~ ($r_m^2 = 0.14$, $r_c^2 = 0.78$) | | | |
| Partner diversity | −0.01 (−0.2, 0.04) | 0.20 | −2.7 | Partner diversity | 0.04 (−0.02, 0.2) | 0.35 | −1.3 |
| Interaction strength | 0.007 (−0.07, 0.2) | 0.20 | −2.8 | Interaction strength | 0.1 (0, 0.3) | 0.60 | 0.80 |
| Migratory distance | 0.02 (−0.05, 0.2) | 0.25 | −2.2 | Migratory distance | −0.003 (−0.1, 0.1) | 0.19 | −2.8 |
| | | | | $x$ | 0 (−0.1, 0.1) | 0.19 | −2.9 |
| | | | | $y$ | 0.3 (0.08, 0.4) | 0.98 | 7.9* |
| | | | | $z$ | −0.02 (−0.2, 0.03) | 0.22 | −2.5 |
| | | | | $a$ | 0.4 (0.3, 0.6) | 1.0 | >15* |
| $y$ ~ ($r_m^2 = 0.015$, $r_c^2 = 0.78$) | | | | Sugar intake ~ ($r_m^2 = 0.091$, $r_c^2 = 0.63$) | | | |
| Partner diversity | −0.1 (−0.3, 0) | 0.61 | 0.87 | Partner diversity | 0.05 (−0.02, 0.3) | 0.37 | −1.0 |
| Interaction strength | 0.008 (−0.1, 0.2) | 0.22 | −2.5 | Interaction strength | −0.03 (−0.3, 0.05) | 0.30 | −1.7 |
| Migratory distance | 0.09 (−0.01, 0.3) | 0.50 | 0.0080 | Migratory distance | −0.002 (−0.1, 0.1) | 0.22 | −2.5 |
| | | | | $x$ | −0.1 (−0.6, 0.02) | 0.50 | −0.036 |
| | | | | $y$ | −0.08 (−0.4, 0.01) | 0.46 | −0.34 |
| | | | | $z$ | −0.07 (−0.5, 0.1) | 0.38 | −0.99 |
| | | | | $a$ | −0.4 (−0.6, −0.2) | 1.0 | 14* |
| $z$ ~ ($r_m^2 = 0.022$, $r_c^2 = 0.60$) | | | | Protein intake ~ ($r_m^2 = 0.041$, $r_c^2 = 0.79$) | | | |
| Partner diversity | 0.001 (−0.09, 0.1) | 0.19 | −2.9 | Partner diversity | −0.03 (−0.3, 0.03) | 0.33 | −1.5 |
| Interaction strength | −0.01 (−0.2, 0.07) | 0.23 | −2.5 | Interaction strength | 0 (−0.1, 0.1) | 0.19 | −2.9 |
| Migratory distance | −0.1 (−0.4, 0) | 0.70 | 1.7 | Migratory distance | −0.002 (−0.1, 0.09) | 0.19 | −2.9 |
| | | | | $x$ | 0.1 (0, 0.5) | 0.53 | 0.25 |
| | | | | $y$ | 0.3 (0.1, 0.5) | 0.99 | 8.6* |
| | | | | $z$ | 0.05 (−0.1, 0.4) | 0.36 | −1.1 |
| | | | | $a$ | 0.02 (−0.04, 0.2) | 0.28 | −1.9 |
| $a$ ~ ($r_m^2 = 0.17$, $r_c^2 = 0.65$) | | | | Anthocyanin intake ~ ($r_m^2 = 0.48$, $r_c^2 = 0.86$) | | | |
| Partner diversity | 0.4 (0.2, 0.5) | 1.0 | >15* | Partner diversity | 0.08 (0, 0.2) | 0.72 | 1.9 |
| Interaction strength | −0.009 (−0.2, 0.07) | 0.22 | −2.5 | Interaction strength | 0.01 (−0.02, 0.1) | 0.23 | −2.5 |
| Migratory distance | −0.3 (−0.5, 0) | 0.97 | 7.1* | Migratory distance | 0.004 (−0.009, 0.07) | 0.13 | −3.7 |
| | | | | $x$ | 0.4 (0.1, 0.6) | 0.99 | 8.6* |
| | | | | $y$ | 0.3 (0.2, 0.5) | 1.0 | >15* |
| | | | | $z$ | 0.6 (0.3, 0.8) | 1.0 | >15* |
| | | | | $a$ | −0.7 (−0.9, −0.6) | 1.0 | >15* |

The structural equation model tested for direct and indirect effects of the partner diversity and interaction strength of frugivores in the networks and their migratory behaviour on the colour profile of consumed fruits (i.e., chromatic colour components ($x$, $y$, $z$) and the brightness ($a$) in avian colour space; see Methods) and on the mean intake of particular nutrients (i.e., lipid, sugar, protein, anthocyanin). The sample size was $n_{obs} = 165$ observations across $n_{species} = 43$ bird species, $n_{site} = 3$ study sites and $n_{time} = 4$ seasons. Animal phylogeny, species, site and season were included as random factors. Given are posterior means of effect estimates (with shrinkage), 95% credible intervals (CI), selection probabilities (P) and $2\log_e$(Bayes factor) (BF) as a measure of support for a given effect. See caption of Table 1 for interpretation of BF-values. Effects that were supported by the variable selection with BF > 2 are shown with an asterisk. The $r^2$ values depict the marginal ($r_m^2$) variance explained by fixed factors only as well as the conditional ($r_c^2$) variance explained by fixed and random factors combined[39]

functional adaptations to inform partner choice and reward regulation may —apart from other selective pressures such as interactions with antagonists[2]—be one driver of the diversification of communicative traits in plant–animal mutualistic networks.

In our study the strength of the relationships between fruit brightness and nutrients was rather low (variance explained by fruit colour after accounting for phylogenetic relatedness of plant species: $r_m^2 = 0.11–0.35$) compared to a previous study about fruit choice of two warbler species in Mediterranean Scrubland ($r^2 = 0.44–0.60$)[5]. The difference in the strength of the colour–reward relationships might be due to geographic variation in the strength of selection by animal mutualists (i.e., geographic selection mosaics)[45] or due to the larger geographic extent of our study. To disentangle the importance of these factors, future studies could test whether geographic variation in the strength of cue–reward relationships in pollination and seed dispersal mutualisms is related to local selection regimes imposed by animal mutualists. Nonetheless, patterns of fruit choice were highly consistent across the three localities in our study with little between-site variance in the brightness profile of selected fruits (variance explained by site: $r^2_{site} = 0.10$; Supplementary Table 2). This suggests that birds are able to regulate their reward intake despite high uncertainty in colour–reward relationships. Because visual discrimination is only the first step of decision-making during foraging[46], animal mutualists may respond to uncertainty in cue–reward relationships by relying on taste, post-ingestive feedbacks or other mechanisms to verify the reliability of cue–reward relationships and adapt their foraging behaviour accordingly[8–11,47,48]. Supporting this idea, previous studies found that flower visiting insects adapt their foraging behaviour in response to intra-individual and inter-individual variation in cue–reward relationships by ceasing interactions with plants whose cues are inaccurate or misleading[10,47]. This may also pose a mechanism for selection on reliability of cue–reward relationships[8,47,49].

The verification of interaction outcomes in resource-based plant–animal mutualisms, such as pollination or seed dispersal, may result in fair trade for two reasons: First, animal mutualists can potentially interact with a range of different partners, which

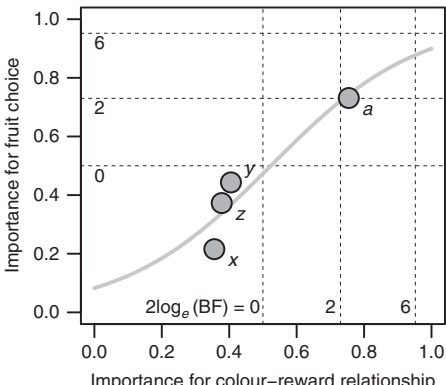

**Fig. 3** Frugivorous birds use the most reliable component of fruit colouration to discriminate the nutrient content of fruit pulp. The marginal selection probability of the chromatic colour components ($x$, $y$, $z$) and the brightness of fruits ($a$) in the avian colour space (see Methods) from the Bayesian model relating fruit colour to the content of nutrients (Table 1) plotted against the marginal selection probability of each colour component from the Bayesian structural equation model relating frugivore traits to the colour profiles of consumed fruits (Table 2). The line represents the fit of a logistic regression model ($y = -2.4 + 4.6x$; $z = 51.8$; $n = 6000$ MCMC samples; $P < 0.001$). Dotted lines indicate $2\log_e$(Bayes factor) (BF). See caption of Table 1 for interpretation of BF-values

creates opportunities for partner choice and punishment (by switching to alternative partners) based on the outcomes of previous interactions. And second, these mutualisms are typically characterized by repeated interactions among multiple partners so that the effect of a single interaction on individual fitness is usually minor compared with the cumulative effects of repeated interactions on fitness[5,50]. Therefore, the self-serving behaviour of animal mutualists may result in fair trade, as long as the cumulative effects of repeated interactions with more rewarding partners offset the negative fitness consequences of a few interactions with less rewarding partners[9,51]. This is in contrast to one-time interaction systems, such as mate choice in semelparous organisms, where a single error of choice can have pronounced fitness consequences for the choosing partner[5,50]. These considerations imply that the reliability that is required to structure partner choice in different systems, as well as the strength of selection on reliability, may depend on how strongly a single interaction affects the lifetime fitness of each partner. We contend that less reliability should be required for communication to structure interactions in systems where partners interact repeatedly and have the option to switch partners based on previous experience. Yet, it is unclear how these conditions might affect the strength of selection on reliability. There is a large body of literature on evolutionary game theory investigating the conditions that allow for the evolution of cooperation in diverse systems[52–55]. However, to our knowledge these models have not yet been used to study under which conditions communication promotes fair trade in plant–animal mutualisms[9] and how system-specific properties (e.g., interaction intimacy or interaction frequency)[56] shape the strength of selection on reliable communication. We suggest that combining recent developments in evolutionary game theory[52–54] with concepts from market and signalling theory[6,57] will likely provide novel insights into the evolution and structural role of communication in diverse biological systems.

Here we integrated concepts from network theory[1], signalling theory[4] and nutritional ecology[15] to assess the role of communication for partner choice and reward regulation in plant–animal mutualistic networks. Our study highlights that partner choice and reward regulation within these networks are determined by

distinct nutritional strategies that are related to species' biology and life history. Importantly, our study provides community-wide evidence that partner choice and reward regulation in resource-based plant–animal mutualisms, such as animal-mediated seed dispersal, require only weak cue–reward relationships. We suggest that weak cues may commonly inform partner choice in plant–animal mutualistic networks, because their exploitation requires only that animals have the sensory and cognitive abilities to verify the reliability of cue–reward relationships during repeated interactions with multiple partners. As this condition is often fulfilled[9,10], our results indicate that communication might be a common mechanism of partner choice and reward regulation in plant–animal mutualistic networks. In a broader context, our results support the idea that, in analogy to human markets, plant–animal mutualistic networks can be considered as biological markets[6] in which consumers rely on advertisement by producers to select those partners whose offer best matches their specific demands.

## Methods

**Plant–frugivore networks.** We compiled ten highly resolved, quantitative plant–frugivore networks from three Central European localities (Fig. 1; Great Britain [GB], Snow and Snow[23]; Germany [DE], Stiebel and Bairlein[58]; Poland [PL], Albrecht et al.[59]). The networks were recorded during focal observations of frugivore visits to plant individuals (GB: 2066 h of observation effort; DE: 1440 h; PL: 2934 h) and describe seasonal interactions (i.e., in spring, summer, autumn and winter) between 44 fleshy-fruited plant species and 48 avian frugivore species, of which we excluded five frugivore species (*Fringillidae* family) that usually consume the seeds and discard the fruit pulp. We used seasonal network representations to account for the fact that potential interactions between plants and frugivores are constrained by their temporal co-occurrence[1]. All studies provided data on the number of feeding visits of each bird species on each plant species per hour (visitation rate per hour hereafter; GB: $n = 20{,}540$ total number of recorded visits; DE: $n = 6353$; PL: $n = 6074$ Supplementary Table 1). Moreover, Stiebel and Bairlein[58] and Albrecht et al.[59] recorded data on the number of fruits consumed per visit for each plant–frugivore pair (fruit consumption rate per visit hereafter)[60]. This data included information for 34 of the 44 plant species and for 39 of the 43 frugivore species in the dataset. For plant–frugivore pairs in GB for which we had data on the fruit consumption rate per visit from DE and PL we used the data from these two localities. When no information about the fruit consumption rate per visit was available for a plant–frugivore pair, we used the mean fruit consumption rate per visit of the frugivore species on other plant species. For four frugivore species without any information on the fruit consumption rate per visit we used the mean value across the remaining 39 frugivore species. We multiplied the visitation rates per hour with the fruit consumption rates per visit to estimate the fruit consumption rate of each frugivore species on each plant species in each network per hour (fruit consumption rate per hour hereafter; GB: $n = 111{,}434$ estimated total number of fruits removed; DE: $n = 31{,}789$; PL: $n = 16{,}366$). No statistical methods were used to predetermine sample size.

To assess the sensitivity of our results to this approach, we conducted the analysis twice, once using networks based on fruit consumption rates per hour and once using networks based on visitation rates per hour. The results of both analyses were virtually identical (Table 2, Supplementary Table 3). This suggests that our conclusions are not affected by the uncertainty associated with missing information on the fruit consumption rate per visit for one of the three localities or by the decision of whether to use fruit consumption or visitation rates per hour as the 'interaction currency'. This is due to the fact that the comparatively large variation in the visitation rates of a frugivore species across different plant species overrides the comparatively small variation in its fruit consumption rate per visit on each plant species[50]. Therefore, the visitation rate of frugivores on plants per hour is a strong predictor of their fruit consumption rates on these plants per hour, regardless of the number of fruits consumed per visit (Supplementary Fig. 1). In the main text we report results based on fruit consumption rates per hour.

Note that we assume the studied plant–frugivore networks to be mutualistic. Therefore, we assume that the fruit consumption rates represent a reasonable first-order approximation of the effect of plants and animals on each other's reproductive performance[50,60,61].

**Fruit traits.** We obtained information about fruit colours (reflectance spectra), fruit morphology (fruit diameter and pulp dry mass) and pulp nutrient content (proportions of lipid, sugar, protein and anthocyanin) for the majority of the 44 plant species in our study from the literature[34,36]. For four plant species contained in the study by Albrecht et al.[59] we did additional measurements of visual, morphological and nutritional fruit traits following the protocols of Valido et al.[36]. We were able to compile data about visual fruit traits and fruit diameter (mm) for all 44 species, data for pulp dry mass (g) for 39 species, data for lipid (g per g fruit

pulp) and sugar content (g per g fruit pulp) for 41 species, protein content (g per g fruit pulp) for 40 species, and anthocyanin content (mg per g fruit pulp) for 32 species.

Our analyses aimed at integrating visual and chemical fruit traits to test for community-wide patterns of fruit choice in frugivores. Because missing data would preclude such a community-wide analysis, we used data augmentation in a Bayesian framework to infer missing data for particular fruit traits during the analysis[62]. A key feature of the models that we use here is that during each iteration of the Markov chain the models predict values for fruit traits of those plant species with missing data based on relationships with explanatory variables and the phylogeny of the plant species (see sections Model 1 and 2 in Statistical analysis). Therefore, the Bayesian framework of our analysis represents a natural way of data augmentation, and allows for complete propagation of the uncertainty that is associated with inferred trait values through the remaining steps of the analysis[62].

**Modelling fruit colours in avian colour space**. Birds have a tetrachromatic colour vision, with four different cone types ranging in their absorption between 300 and 700 nm[63]. Therefore, we measured the reflectance spectra of fruits in 5-nm-wide intervals over the range of 300–700 nm[36]. To calculate fruit colours according to the visual perception of birds, we transformed the fruit reflectance spectra into relative cone excitations of the four cone types[63]. Then we transformed the relative cone excitation values of the four avian cone types into the tetrahedral colour space of birds[37,38], where each of the vertices represents the sole excitation of a single cone. The colour space within the tetrahedron is characterized by three Cartesian coordinates ($x$, $y$, $z$) that are related to particular chromatic colour components and define the location of each spectrum[37,38]. The origin of the coordinates is the achromatic point where all cones are stimulated equally. Chromaticity, or the strength of a colour signal, is proportional to the Euclidean distance from any point within the tetrahedron to the achromatic point[37]. The $x$-coordinates range from blue (negative scores) to red (positive scores), $y$-coordinates range from purple (negative scores) to green (positive scores), whereas $z$-coordinates indicate UV reflectance (positive scores). We quantified perception of fruit brightness based on the excitation of the avian double cone using achromatic values ($a$)[64]. High achromatic values indicate that a fruit is perceived as being bright, whereas low values indicate that a fruit is perceived as being dark.

In general, the visual system of birds is relatively conservative. The spectral sensitivities of most frugivorous birds are unknown, but most passerine families belong to the ultraviolet sensitive (UVS) type of colour vision, where the sensitivity of the short-wavelength cone is biased towards the ultraviolet. We therefore used the well-known UVS spectral sensitivities of blue tits (*Cyanistes caeruleus*) to model fruit colour perception[65].

**Frugivore traits**. To test the hypothesis that the mean colour profile of consumed fruits and the mean nutrient intake of frugivorous birds are related to their roles in the networks and to their migratory behaviour we used three measures: (i) the diversity of plant species consumed by a frugivore species relative to the number of plant species available in a network (partner diversity), (ii) the relative contribution of a frugivore species to fruit removal in a network (interaction strength), and (iii) the latitudinal distance between the study sites and the wintering range of a frugivore species (migratory behaviour). More precisely, we quantified partner diversity as the effective number of plant species consumed by a frugivore species in a given network based on the exponent of the Shannon diversity of links ($e^H$)[66]. To account for variability in plant diversity between networks, we standardized the effective number of consumed plant species by the number of available plant species in a given network. Therefore, our measure of partner diversity quantifies relative niche breadth[67]. We calculated interaction strength as the sum of fruits removed by a frugivore species in a network per hour divided by the sum of fruits removed by all frugivore species in that network per hour. Interaction strength, thus, quantifies the importance of a frugivore species in terms of its relative contribution to the community-wide rate of seed dispersal in a network[50]. We quantified the migratory behaviour of birds as the latitudinal distance between the locations of the study sites, which are situated in the breeding ranges of the birds, and the centroids of the wintering ranges extracted from distribution maps[68].

**Plant and frugivore phylogenies**. We extracted information about the phylogenetic relatedness of plants from a dated phylogeny of a large European flora[69] (Fig. 1b). For birds we obtained a random sample of 1000 phylogenetic trees from www.birdtree.org[70], and calculated a maximum clade credibility tree with median node heights (Fig. 1c).

**Visual and nutritional profile of plant–frugivore interactions**. We integrated the visual, nutritional and morphological fruit traits in the context of community-wide interactions between plants and frugivores. To calculate the mean colour profile $\bar{c}_j$ of the fruits that frugivore species $j$ consumed in each of the ten networks, we estimated the mean of the species-specific colour coordinates ($x$, $y$, $z$) and brightness ($a$) of all plant species consumed by frugivore $j$ in a given network

weighted in proportion to their interaction frequency in that network as:

$$\bar{c}_j = \sum_{i=1}^{I} c_i f_{ij} \Big/ \sum_{i=1}^{I} f_{ij}, \tag{1}$$

where $c_i$ is the colour value of plant species $i$, $f_{ij}$ is the interaction frequency between plant species $i$ and frugivore species $j$ in a given network, and $I$ is the total number of plant species in the networks. To calculate the mean nutritional profile $\bar{n}_j$ of the fruits that frugivore species $j$ consumed in each of the ten networks, we estimated the mean intake of nutrients (i.e., lipid, sugar, protein and anthocyanin) per gram dry pulp mass as:

$$\bar{n}_j = \sum_{i=1}^{I} n_i m_i f_{ij} \Big/ \sum_{i=1}^{I} m_i f_{ij}, \tag{2}$$

where $n_i$ is the proportion of nutrients per gram dry pulp mass, $m_i$ is the pulp dry mass per fruit of plant species $i$, and $f_{ij}$ is the interaction frequency as above.

**Statistical analyses**. We used a Bayesian hierarchical model to test our main hypotheses that (i) fruit colour is an indicator of nutritional rewards, and that (ii) fruit choice by frugivores is mediated by fruit colour and related to frugivore traits. In brief, the Bayesian hierarchical model that we developed contained three distinct sub-models, which integrated the information about the traits and phylogenetic relationships of plants and animals in the context of their interactions in a community-wide network analysis. The models that we used here are reviewed in Nakagawa and Santos[71]. The most general form of the models can be written as:

$$\mu_i = \eta_i + \zeta_i, \tag{3}$$

$$y_i \sim \text{Normal}(\mu_i, \sigma_\varepsilon^2). \tag{4}$$

where $\mu_i$ is the expected value of the $i$th observation of response variable $y$, which follows a normal distribution around $\mu_i$ with residual variance $\sigma_\varepsilon^2$ (Eqs. 3 and 4). The design component $\eta$ contains an intercept $\alpha$, a matrix with fixed effects $\mathbf{X}$ and an associated vector of parameters $\theta$:

$$\eta_i = \alpha + \sum_j \theta_j \mathbf{X}_{ij}. \tag{5}$$

The component $\zeta$ contains random effects associated with species' phylogeny ($a_k$), species' identity ($v_k$), sites ($s_l$) and seasons ($t_m$):

$$\zeta_i = a_{k[i]} + v_{k[i]} + s_{l[i]} + t_{m[i]}. \tag{6}$$

In the simplest form, when only one observation per species enters the model, the models only contain a phylogenetic random effect, $a_k$, for the $k$th species, where $\mathbf{a}$ is a 1-by-$N_{\text{species}}$ vector of $a_k$, which is multivariate-normally distributed around 0, $\sigma_a^2$ is phylogenetic variance, and $\mathbf{A}$ is the inverse of a $N_{\text{species}}$-by-$N_{\text{species}}$ correlation matrix of distances between species, extracted from a phylogenetic tree[71]:

$$\mathbf{a} \sim \text{Normal}(0, \sigma_a^2 \mathbf{A}) \tag{7}$$

If more than one observation per species is available; the models have multiple levels containing additional random effects for species ($v_k$, which is estimated in addition to $a_k$), sites ($s_l$) and seasons ($t_m$):

$$\mathbf{v} \sim \text{Normal}(0, \sigma_v^2 \mathbf{I}), \tag{8}$$

$$\mathbf{s} \sim \text{Normal}(0, \sigma_s^2 \mathbf{I}), \tag{9}$$

$$\mathbf{t} \sim \text{Normal}(0, \sigma_t^2 \mathbf{I}), \tag{10}$$

where $\mathbf{v}$ is a 1-by-$N_{\text{species}}$ vector of $v_k$, which is normally distributed around 0 with species-specific variance $\sigma_v^2$, $\mathbf{s}$ is a 1-by-$N_{\text{site}}$ vector of $s_l$, which is normally distributed around 0 with site-specific variance $\sigma_s^2$, $\mathbf{t}$ is a 1-by-$N_{\text{time}}$ vector of $t_m$, which is normally distributed around 0 with season specific variance $\sigma_t^2$, and $\mathbf{I}$ is an identity matrix.

To identify the most informative variables we used a Bayesian indicator variable selection with global adaptation[72]. Indicator variable selection combines the effect size $\beta_j$ with an indicator variable $I_j$ to denote whether the regression parameters $\theta_j$ are in the model or not (where $I_j = 1$ indicates presence, and $I_j = 0$ absence of covariate $j$ in the model). Then we set $\theta_j = I_j \beta_j$ assuming that the indicators and effects are independent a priori, so $P(I_j, \beta_j) = P(I_j) P(\beta_j)$, and independent

priors are placed on each $I_j$ and $\beta_j$:

$$P\left(I_j = 1\right) \sim \text{Bernoulli}(0.5), \tag{11}$$

$$\beta_j \Big| \left(I_j = 1\right) \sim \text{Normal}\left(0, \sigma_\beta^2\right), \tag{12}$$

$$\sigma_\beta^2 \sim \text{Uniform}(0, 100), \tag{13}$$

where the prior inclusion probability was set to 0.5, and the variance $\sigma_\beta^2$ was estimated by the model. We used a uniform prior between 0 and 100 for $\sigma_\beta^2$. This form of global adaptation has the advantage of facilitating the tuning of the variable selection, because the distribution of each $\theta_j$ is shrunk towards the correct region of the parameter space by the other parameters in the vector of regression coefficients $\theta$ in the model. We used $2\log_e(\text{Bayes factor})$ as a measure of evidence for a given effect (BF hereafter)[73]. Values of BF < 2 indicate no support; values between 2 and 6 indicate positive support; values between 6 and 10 indicate strong support; and values > 10 indicate decisive support. To assess model fit we used the marginal variance ($r_m^2$) that is explained by the fixed factors, as well as the conditional variance ($r_c^2$) that is explained by the fixed and random factors combined[39]. For sub-model 3, we also quantified the amount of the variance in fruit choice that is explained by each of the random factors separately (Supplementary Table 2).

In sub-model 1, we fitted the relationship between pulp dry mass and fruit diameter of each plant species to infer pulp dry mass for those plant species with missing data. Later we used the pulp dry mass for calculating the weighted mean nutritional profiles of the fruits that a frugivore species consumed (see Eq. 2). The model contained a phylogenetic random factor to account for the phylogenetic relatedness of plants. We transformed pulp dry mass and fruit diameter to their natural logarithm before analysis. Conditional $r^2$-values indicated that the model was able to infer pulp dry mass for species with missing data with high accuracy ($r_c^2 = 0.82$; Fig. 1d).

In sub-model 2, we tested our first hypothesis that fruit colour is a reliable indicator of nutritional rewards. To do so, we fitted the nutritional contents of each plant species (i.e., lipid, sugar, protein and anthocyanin) as response variables and the colour coordinates ($x$, $y$, $z$) and brightness ($a$) of fruits in the avian colour space as explanatory variables. Similar to sub-model 1, this model contained a phylogenetic random factor to account for the phylogenetic relatedness of plants. We transformed the lipid, sugar, protein and anthocyanin content of fruits to their natural logarithm before analysis. Conditional $r^2$-values indicated that the model was able to infer the nutrient content of plant species with missing observations with low to moderate accuracy (lipid: $r_c^2 = 0.51$; sugar: $r_c^2 = 0.23$; protein: $r_c^2 = 0.16$; anthocyanin: $r_c^2 = 0.42$; Fig. 1d).

In sub-model 3, we tested the second hypothesis that the mean nutrient intake of frugivorous birds is related to the partner diversity and interaction strength in the networks and to their migratory behaviour and that it is primarily mediated by the mean colour profiles of consumed fruits (i.e., frugivore traits are indirectly related to nutrient intake via the colour profiles of consumed fruits; see Fig. 2d). To do so, we used a Bayesian structural equation model to test whether the partner diversity and interaction strength of frugivores and their migratory distance were directly related to the mean lipid, carbohydrate, protein and anthocyanin intake of birds, or indirectly mediated by the mean colour coordinates ($x$, $y$, $z$) and the mean brightness ($a$) of consumed fruits (Fig. 2d). The structural equation model contained a phylogenetic random factor for birds, as well as random factors for bird species, site, and season. We transformed the response variables mean brightness of consumed fruits, mean lipid, sugar, protein, and anthocyanin intake, as well as the predictor variables partner diversity and interaction strength to their natural logarithm before analysis. In the model, we explicitly considered potential seasonal changes in foraging behaviour of migratory birds by setting their migratory distance outside the pre-migration and migration periods to zero (i.e., during winter and spring). Therefore, we assume that during the breeding season migratory and resident birds exhibit similar fruit preferences. In addition, we also considered two alternative models. One model that included the partner diversity and interaction strength of frugivores and their migratory distance, but without setting the migratory distance outside the pre-migration and migration periods to zero (alternative model 1); and a second model that included the partner diversity and interaction strength of frugivores, their migratory distance, period (migration versus non-migration) and the interaction between migratory distance and period as fixed factors (alternative model 2). These two alternative models yielded identical conclusions regarding the fruit choice of resident and migratory birds (Table 2, Supplementary Tables 4 and 5). However, alternative model 1 had lower explanatory power ($r_m^2 = 0.12$) than the model in which we set the migratory distance outside the pre-migration and migration periods to zero ($r_m^2 = 0.17$; Table 2 and Supplementary Table 4). Moreover, the inclusion of period and its interaction with migratory distance in alternative model 2 did not improve the explanatory power of the model ($r_m^2 = 0.18$; Table 2 and Supplementary Table 5). Therefore, we report the results of the simpler model in which we set the migratory distance of migrants during the non-migration period to zero in the main text.

Finally, we aimed to assess whether birds used the most reliable component of fruit colouration to discriminate the nutrient content of fruit pulp and to regulate their nutrient intake. To do so, we quantified for each colour component how often it was selected as predictor of nutrient content in fruit pulp across the plant species (sub-model 1; Table 1), and how often it was selected as being related to the birds' partner diversity and interaction strength in the networks and to their migratory behaviour (sub-model 2; Table 2). Therefore, we determined the overall importance of the colour components for colour–reward relationships based on marginal selection probabilities (i.e., based on the mean selection probability of each colour component across response variables in sub-model 1). Likewise, we determined the overall importance of colour components for fruit choice based on marginal selection probabilities of the paths relating frugivore traits to the colour profile of consumed fruits (i.e., based on the mean selection probability of each colour component across predictor variables in sub-model 2). The former gives an indication of the relative importance of a colour component for colour–reward relationships, the latter provides information about the relative importance of a colour component as mediator of fruit choice by birds. We used a logistic regression to test whether the selection probabilities of the colour components in sub-model 2 are positively related to their selection probabilities in sub-model 1.

The model was implemented in JAGS[74], and run in $R$[75] through the *rjags* package[76]. The JAGS code for the data analysis is given as part of the Supplementary Material (see Supplementary Data 1). We ran eight parallel chains for the model. We used uninformative priors for all parameters and the initial values were drawn randomly from uniform distributions. Each chain was run for 26,000 iterations with an adaptive burn-in phase of 1000 iterations and a thinning interval of 100 iterations, resulting in 250 samples per chain, and 2000 samples from the posterior distribution. We checked the chains for convergence, temporal autocorrelation, and effective sample size using the *coda* package[77] (Supplementary Tables 6 and 7). We checked residuals for normality and variance homogeneity.

**Code availability**. The computer code of the analyses is available in figshare with the identifier https://doi.org/10.6084/m9.figshare.674068[78]. The JAGS code for the Bayesian hierarchical model is also given as part of the Supplementary Materials (Supplementary Data 1).

**Reporting Summary**. Further information on research design is available in the Nature Research Reporting Summary linked to this article.

## Data availability
The data that support the findings of this study are available in figshare with the identifier https://doi.org/10.6084/m9.figshare.674068[78]. A reporting summary for this article is available as a Supplementary Information file.

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

## Acknowledgements

J.A., D.G.S. and N.F. thank the administration of the Białowieża National Park, the forestry administrations of Białowieża, Hajnówka and Browsk, and Polish authorities (Ministry of Environment, GDOS and RDOS) for the permissions to work in Białowieża Forest. J.A. was supported by the German Federal Foundation for Environment (DBU) and by the German Academic Exchange Service in the framework of a post doctorate fellowship grant (DAAD, No. 91568794).

## Author contributions

J.A., J.H., H.M.S. and N.F. conceived the study. J.A., J.H., D.G.S. and H.M.S. collected data. J.A. and J.H. conducted analyses and J.A. wrote the first draft of the manuscript with input from J.H. All authors discussed the results and contributed to revisions of the manuscript.

## Additional information

**Competing interests:** The authors declare no competing interests.

