## [Peer Review File · Nature Communications]

Reviewers' Comments:

Reviewer #1:

Remarks to the Author:

This paper examines fruit choice in mutualistic interactions among frugivore birds and fruit-producing plants. The authors show that fruit brightness, but not chromatic (colour) features, are indicative of nutrition content (brighter = higher lipid, darker = higher sugar and anthocyanin), and then show that non-migratory and migratory birds, and generalist and specialist foragers, optimise their fruit choices differently: non-migratory and generalist birds select dark fruits and migratory and specialist birds select bright fruits. This matches expected physiological needs of non-migratory/migratory and generalist/specialist birds.

The data analysed comes mostly from the published literature: three studies on frugivore mutualism networks and two on fruit nutritional content; additional measurements were taken for four fruit species.

The authors emphasise that theoretically their work is the first to show on a community scale that weak associations between fruit colour and nutrition are capable of resulting in strategies that appropriately regulate birds' nutritional intake.

Before going further, I note that I write this review from the perspective of being an evolutionary biologist with expertise in network science and Bayesian statistics, with background in avian behaviour, but without specific expertise in mutualism or foraging. Thus my comments on the theoretical aspects of mutualism are coming from the perspective of an interested non-expert.

My largest issue with the paper is the emphasis on the notion that weak associations can lead to optimised intake. This may be due to my inexpert status with respect to this theory, but in any case, I do not see in the paper how the associations shown here between brightness and nutrition value are weak. Is it because it is brightness only and not chromatic value? Is it statistically weak? In comparison to what? I am somewhat wary of arguments made on the basis of relative strength of statistical significance, but if it was presented with good rationale, I might be convinced. What would be a 'strong association'? A reference to an opposite association shown in another location is noted in the discussion – is this relevant to the weakness? Something to enable the inexpert reader to understand how the results here are evidence of a weak association is needed.

Related to the above, I feel that the non-expert reader simply needs to be provided a little bit more framework surrounding the import of this work. I learn from the text that this is the first time this has been shown, and that the issue is "highly contentious and unresolved" – but I do not get a good sense of what is at stake, thus making it difficult to assess the impact of this work for wider evolutionary biology. What are the implications of the findings here? Perhaps even more interesting, what would be the implications of weak associations being incapable of leading to optimised intake – the hypothesis for which a counter example is now presented – e.g., what has been ruled out? How has this changed the understanding of the evolutionary trajectories of mutualistic networks? The authors need to more strongly state the case, and explain to the practitioners, of this work's interest to the wider field.

Beyond the concern above, the paper presented a (mostly) clear and interesting read, and is an impressive use of multiple different data sets to inform theory in a novel way. The methodology appears rigorous and I am very happy (with my Bayesian hat on) with the statistical analyses. The data augmentation was described nicely and looks robust. I also very much appreciate the code for the analysis being provided in supplementary material.

Other major comments:

1. While the writing style is clear and straightforward, and the material mostly easily digestible, throughout the paper there is a pattern of presenting material slightly back-to-front, such that specific terms are used and then a few sentences later, their meaning is clarified (or occasionally only in the Methods/Supplementary Material). I would suggest the authors go through the paper with an eye to this, and make sure that any terms or references to things like methodology are actually understandable by the naïve reader at the moment of presentation. Please be aware that a reader of the journal is unlikely to flip to the methods or supplementary material each time one is mentioned: thus the text needs to be understandable at first read without the additional details found there. A few examples:

a. p6 line 128: "the interaction strength of birds" is only defined in the Methods and has not been presented before this point even in vague terms as a feature of interest (perhaps because it turns out to have no association, but if it is tested, some rationale as to what it is and why it might be theoretically of interest should be provided).

b. p11-12 section "Frugivore traits": this section follows the pattern of presenting undefined terms which are explained slightly later. Only subtle rewording would be required to fix such things, for example instead of "namely (i) the interaction strength, (ii) the proportional generalization (generalization hereafter)" I suggest something like "namely (i) the relative importance of a frugivore within the mutualism network, its interaction strength, (ii) a proportional measure of its dietary generalization (generalization hereafter)". The description of both interaction strength and generalization brings the reader through a number of details before summarizing what the measure actually captures: better to start with this, so that the details make sense. It was only at the very end of the migratory distance part that I concluded that the study sites must be on breeding grounds; this should be clear earlier.

c. p13 section "Modelling fruit colours...": the word 'Achromatic' starts the sentence preceding the one that defines it: "There, we use the term 'brightness' when referring to 'achromatic values'"; it ought to be defined first. Additionally, how "perception of fruit brightness" is measured is described right up in the front of this paragraph – I suggest introducing the term 'achromatic' here rather than after the discussion of it is over.

d. p14 line 14: referring to an equation several paragraphs before the equation leads to confusion – can the "Quantification" section be put before the Statistical analyses rather than interleaved into the submodels?

e. p17 and Fig 3b: it took quite a lot of thinking to figure out that the y-axis in figure 3b must represent 1-the proportion of colour profiles "inside the 95% confidence interval of the simulated colour profiles" (lines 411-412), because "non-random" must mean they are *outside* these confidence intervals. This whole analysis could be presented much more clearly.

f. Supplementary Information p2: a selection of variables is presented on line 30 but not defined until line 38 – these could be moved closer (or inverted).

Additional minor comments:

2. p5 line 95: suggest saying "An achromatic component" rather than "The achromatic component" since using the definite article suggests this is something or a component of something you've discussed already, yet it is an additional component beyond the tetrahedral shape defined by the chromatic components. (I spent a brief period of time trying to understand what particular chromatic colour component "The achromatic component" could possibly be, before continuing to read and having my wonder clarified: this is a very subtle form of the issue noted in comment 1, above.)

3. p5 line 98 & p6 line 113: the fact that there is stochastic variable selection appears only in the

supplementary material; a reader who looks through the methods for this phrase will be disappointed. I suggest referring to both Methods and Supplementary Methods here.

4. p11 line 254-255: I highly suspect that “the interaction frequency of each frugivore species” is meant to be something like “the number of interacting plants for each frugivore speices” – as frequency already means it’s divided by something! If not, where this frequency comes from needs to also be defined.

5. p15 equation (1): the variable J, the total number of frugivore species in the network, should be explicitly defined somewhere in the text.

6. p16 lines 388-398: Could you please provide the outcome of these checks in the Supplementary Materials.

7. p17 line 413: should “differed between” be instead “differed among”? Unless the test was comparing the entirety of the chromatic components to brightness, but I don’t think this is the case. (And not pre-supposing your answer would be better there).

8. Tables and Figures: throughout the tables and figures, chromatic components are referred to in text as lower case letters {x,y,z} but in the table/figure as upper case X, Y, Z (with the exception of the axes labels in Fig 2a,b). These should be made congruent in some manner – either with explanatory text (clunkier) or simply using upper or lowercase everywhere.

9. p26 Table 2 title: the term “phylogenetic path analysis” is not present anywhere else in the text, and while it can be figured out by reading in the Methods that the Bayesian hierarchical model is a path analysis and includes phylogeny as a random effect, at the time of reference to Table 2 the naïve reader has only seen a “Bayesian hierarchical model with a stochastic variable selection procedure” referred to: better to use a term more related to this.

10. p29 line 648: Achromatic values {A} are referred to in reference to panels a and b, and yet they do not appear to be present in these panels? Either remove this reference or it might actually be possible to represent A as shades of grey in the plots?

11. p31 Fig 3a: Y and Z overlap in the presented plot. Can the labels to the dots be moved so they do not overlap?

12. Supplementary Information p3 description of equations 6-8: that I represents the identity matrix should be stated (assuming it does).

13. Supplementary Information p8 Supplementary Table 1: Many undescribed abbreviations are presented; for example, the studies PL, DE, and GB should be re-identified here so this can be referred to in isolation from the main text, and what do all the letters in the season column mean? (Oh, perhaps those are months?).

14. Supplementary Information p8 Supplementary Table 1: I would much prefer if the actual data matrices tabulated could be provided in addition this summarizing information.

Reviewer #2:

Remarks to the Author:

General Comments

The authors address whether fruit brightness provides a reliable cue to birds about the nutritional quality of the fruits across several different ecosystems, and whether birds select fruits based on certain fruit traits. One of the more valuable components of this work (and a primary goal of this study) is the analysis across several plant-frugivore systems of the relationship between fruit colour and nutrient composition of these fruits. Their novel conclusion is that the plant-frugivore mutualism can be maintained with only weak correspondence between the cue (fruit brightness) and the reward (nutritional quality). One of the less valuable components of this work (and the second goal of this study) is the attempt to relate fruit selection by frugivores to these same fruit characteristics. This latter goal falls short primarily because (a) the rationale provided is weak, and (b) the datasets used for evaluating this goal are incomplete and required too much estimation. These points are described below in more detail.

The rationale for the work is quite broad (I would say too much so): mutualistic networks (they argue that the plant-frugivore interaction is mutualistic and thereby benefits the plant and frugivore) are maintained by reliable communication (in this case, fruit brightness communicates nutritional reward to the frugivore) between the partner mutualists. The conceptual problem with arguing that the system is a mutualistic network is in part related to the authors' comparison across the four seasons between resident birds (that they suggest select bright, lipid-rich fruits) and migratory birds (that they suggest select sugar- and antioxidant-rich fruits) for which the mutualistic relationship might be quite distinct. This suggests that the fruits selected by resident birds would be distinct from that of migrating birds which also means that the pair-wise mutualisms would also be distinct. However, the authors recognize that migratory birds might select different fruits across the four seasons, so they assume that migrants and residents have similar fruit preferences during the breeding season. How such seasonal changes in bird abundance, fruit phenology and abundance, and fruit selection results in true mutualisms is not well established in this manuscript. In sum, the reliance on a rationale related to mutualistic networks is weak and does not do justice to the interesting results.

The three objectives outlined by the authors (lines 77-82) differ in their merit and are not consistently outlined throughout the manuscript. The first objective (assess the relationship between fruit colour and fruit nutrient composition) is well worth doing in part because this is done on a refreshingly broad scale (across 44 plant and 43 bird species). The second objective is confusing (not sure what this means: "preferences for particular nutrients are related to their generalization on fruit resources".) The third objective is trivial or too general – no vertebrate animal is expected to feed randomly, and testing whether birds "optimize their nutritional rewards" is not adequately informative (i.e., there is no discussion about the optimization criteria or how the fruit traits might translate into true 'rewards' for the frugivore – the latter requires knowing what nutrients are assimilated relative to the requirements of the birds). In fact, the 2nd and 3rd objectives are better described in the Stats Analyses section (lines 319-321): the 2nd hypothesis is that fruit choice by frugivores is mediated by fruit colour and related to frugivore traits. Although later in the methods this becomes hypotheses 2 and 3 (lines 366-367): fruit color mediates fruit choice and is related to frugivore traits. These latter statements of hypotheses (whether one or two) are a better characterization of what was done. In sum, the readers' enthusiasm for the work waned by the end of the introduction because of the weak rationale, the heavy jargon (see Specific Comments below), and the lack of compelling objectives (except for the first one).

The main results take advantage of previously compiled datasets on fruit characteristics, bird and plant phylogenies, bird vision capacity, and plant-frugivore visitation and consumption rates. The original results are the product of applying these previously compiled datasets to a meta-analysis of sorts that used a Bayesian hierarchical modeling framework. The completeness of these compiled

datasets seemed adequate for testing hypothesis #1 (fruit colour related to nutrient composition) and found these results quite compelling. Unfortunately, the completeness of the datasets was not adequate for testing the other (or two) hypothesis (fruit choice related to fruit colour or nutrients). Most importantly, plant abundance data was only available for two of the ten networks (line 398) and so fruit consumption information was used to estimate plant abundance – this confounds availability and use information which is the basis for testing this 2nd (and 3rd) hypothesis. This undermines the reader's confidence in the network analyses on this point.

Specific Comments

Lines 32-44: the jargon is quite heavy in this first paragraph to the point of confusion in several places. For example (#1), the statement "a fundamental question is whether communication among mutualists serves to attract partners via increased conspicuousness, or whether it also informs partner choice and reward optimization" does not adequately make clear to the general reader why these two parts (conspicuousness vs. reward optimization) are mutually exclusive, and what exactly is meant by "partner choice" and "reward optimization". For example (#2), this statement is quite dizzying and confusing even after reviewing the citation: "in most mutualisms individual fitness does not depend on the outcome of single erroneous interactions, but on the summed effect of repeated interactions" – needs better context.

Line 56: why would it be dietary preferences OR migratory behaviour? (birds during migration also exhibit strong diet preferences). The authors should also be careful about the word choice here: diet selection (use vs. availability) is distinct from diet preference (use given equal availability).

Lines 59-61: this is an oversimplification. Birds (or most any other vertebrate) do not usually or commonly compensate for a protein deficit during migration or at other times of the annual cycle. The best examples come from the many, many studies of birds and mammals where energy density of diet is kept constant and the protein concentration is reduced (usually as carbohydrate is increased to keep the energy density constant) – these are the classic experiments where minimum protein requirements of an animal are defined. In such studies, animals usually eat the same amount of food on all diets (since they are isocaloric) but since the protein levels eventually are insufficient as dietary protein levels decline, there are signs of deficiency that allow the quantification of minimum protein requirements. Any animal nutrition textbook outlines these more general patterns – my favorite for birds is Kirk Klasing's "Comparative Avian Nutrition" book (CAB International, UK; 1998).

Lines 66-73: these predictions about fruit traits (e.g., fat, carbohydrates, antioxidants) and their selection by resident vs. migrating birds is also oversimplified. Both groups of birds are likely maximizing their energy intake (not just resident birds as the authors argue), and all birds can fatten on fruits that are either high-carbohydrate or high-fat (birds that eat the former just need to use de novo lipogenesis to produce body fat). This undermines the primary rationale for comparing fruit selection in resident and migratory birds in this study.

Lines 145-147: this statement about the main novelty of this study is telling. I agree that the authors have shown weak associations between fruit colour and nutrient composition of fruits. I find little compelling evidence for the latter claim: frugivores "regulate their reward intake according to specific nutritional strategies". The next few sentences talk about "reward optimization", "mutualistic networks", "partner choice", "functional adaptations" all of which are assumed and undocumented.

Reviewer #3:

Remarks to the Author:

Albrecht et al present a well-framed and written manuscript describing how weak cues from fruits drive frugivore selection and how this is related to the optimization of nutritional rewards. The authors compiled a large amount of high-quality data and they used the state of the art in statistical analyses to answer a very interesting question. In general, I greatly enjoyed the study and I think it may be of the interest of many other ecologists working in a wide range of fields. Also, except for some small questions on the methods (see below), the authors have included enough information to reproduce the study (given that they make the raw data available). I am describing further suggestions and questions related to specific lines:

Main text

- Title: The first time I read it I was not sure about what would be the content of the manuscript. I can totally understand that authors want to make the title and the study as much generalizable as possible, but in this case the main issue of the study gets lost. I would suggest other more informative titles such as (but not limited to):

- o Reward optimization in mutualistic networks requires weak cue-reward relationships

- o Weak cues inform partner choice and reward optimization in mutualistic networks

- Abstract, page 2, line 27 (P2L27): As written I thought you were going to show physiological experiments in the study. I'd rephrase to be clear that you are talking about previous studies. Maybe something like "consistent with previous physiological experiments".

- P2L28-30: The authors write here and repeat several times in the text: "the extraordinary diversity of communicative traits in mutualistic networks originates from functional adaptations to inform partner choice and reward optimization", also in P7L150-151 and in P10, L222-225: Even though this is a plausible explanation for the pattern, I don't think their results demonstrate that. Several forces may be driving diversification in mutualist species communicative skills and their statement sound like an over-interpretation of their results. This may be suggested in the discussion as a possibility, but I would try to soften the affirmation and avoid including it in the abstract.

- P3L35-36: Related to the previous comment, why is this the only option? Why cannot this be at least partially caused by not other processes?

- P3L47: "The composition of fleshy fruits is often imbalanced..." what kind of imbalance are you referring to? In relation to what? Please be more specific, like it is written is ambiguous.

- P8L164-168: In this paragraph you say that the evolution of generalist mutualists may require both behavioral (...) adaptations and also "specific nutritional strategies that enable animal mutualists to rely solely on resources that they acquire through mutualistic interactions". To be, this nutritional strategies are a kind of behavior. Maybe you should delete "behavioural" from line 166.

- P9. In lines 189-192 you underline that cue-reward relationships can change geographically based on a study by Schaefer et al. (2014), and then in lines 207-210 you say that communication may be "remarkably robust to spatiotemporal variation". This seems a bit contradictory to me. This made me think about whether the results were consistent among your 3 study areas. In the analyses you included study area as a random factor. I agree with this, but this accounted for the between sites variability and we cannot really see how things changed among sites. I would like to see whether indeed results are similar for the different study areas. Maybe you can show the statistics for the random factor or do a simpler analysis where study area is included as a fixed factor just to see its effect. This will allow you to make more robust affirmations about the spatial variability.

Results

- P11L241: How were visitation rates calculated? Number of visits/hour? Please be specific, it is easy to get lost with so many calculations.

- P11L241-244: If, as you say in the last sentence of P10 and the first one of P11, you only compiled

information from DE and PL about the number of fruits consumed/visit, how did you calculate the rate of fruit removal for the GB data?

- P15 equations 1 & 2: I think that there may be either something missing in the equation or in the description. Equation 1 & 2 seem to give the color and nutritional profile of a specific fruit i , and not the "mean color/nutritional profile of the fruits that bird j consumed". I think the equation needs to be completed to express the mean of all the plants.

- P15L363: I guess the "pulp dry mass of plant species" is the pulp dry mass per fruit of each plant species?

- P17L409. I think the n_j should be c_j here.

- Page 17: In your randomization approach, I am wondering how good was the estimation of the relative availability of the plant species. Maybe you can compare the results from the simulation with the data for the two study sites for which you had abundance data. Are they highly correlated? Alternatively, you can repeat the analysis using only the two sites for which you had data on abundance and see if the results are consistent.

- P17L412: I am not sure what do you mean for "(random: true versus false)" and (interaction term: random x color component), can you please explain it a bit more?

- Figure 1. Very nice and informative figure! I'd also describe in the caption or in the figure what x, y, z and A are so that reader do not need to go to the text, and also what GB, DE and PL stand for.

- Figure 3b: I think this figure needs to be explained a bit better, I am still not 100% sure of what it is representing, even though the results described in lines 138-142 are clear.

We thank the Reviewers for their constructive and positive comments, which in our opinion helped to improve the manuscript. Below we provide a point-by-point response to the comments, which are highlighted in red font. To facilitate the review process, we have also highlighted the changes made to the manuscript in red font in the Word document. Please note that line numbers in our responses refer to lines in the revised version of the manuscript.

Reviewers' comments:

Reviewer #1 (Remarks to the Author):

This paper examines fruit choice in mutualistic interactions among frugivore birds and fruit-producing plants. The authors show that fruit brightness, but not chromatic (colour) features, are indicative of nutrition content (brighter = higher lipid, darker = higher sugar and anthocyanin), and then show that non-migratory and migratory birds, and generalist and specialist foragers, optimise their fruit choices differently: non-migratory and generalist birds select dark fruits and migratory and specialist birds select bright fruits. This matches expected physiological needs of non-migratory/migratory and generalist/specialist birds.

The data analysed comes mostly from the published literature: three studies on frugivore mutualism networks and two on fruit nutritional content; additional measurements were taken for four fruit species.

The authors emphasise that theoretically their work is the first to show on a community scale that weak associations between fruit colour and nutrition are capable of resulting in strategies that appropriately regulate birds' nutritional intake.

Before going further, I note that I write this review from the perspective of being an evolutionary biologist with expertise in network science and Bayesian statistics, with background in avian behaviour, but without specific expertise in mutualism or foraging. Thus my comments on the theoretical aspects of mutualism are coming from the perspective of an interested non-expert.

My largest issue with the paper is the emphasis on the notion that weak associations can lead to optimised intake. This may be due to my inexpert status with respect to this theory, but in any case, I do not see in the paper how the associations shown here between brightness and

nutrition value are weak. Is it because it is brightness only and not chromatic value? Is it statistically weak? In comparison to what? I am somewhat wary of arguments made on the basis of relative strength of statistical significance, but if it was presented with good rationale, I might be convinced. What would be a ‘strong association’? A reference to an opposite association shown in another location is noted in the discussion – is this relevant to the weakness? Something to enable the inexpert reader to understand how the results here are evidence of a weak association is needed.

Response: Thank you for bringing this lack of clarity to our attention. When we refer to weak association we mean the covariance between fruit nutrients and fruit colour in terms of the marginal r^2 -values from the regression models presented in Table 1. Therefore, by ‘weak’ we mean ‘low statistical predictability’ of fruit nutrients from fruit colouration after accounting for phylogenetic relatedness of the plants. We added this information to the first paragraph of the results section (Lines 126–128) and also provide the range of r^2 -values in the respective sentence of the abstract where we refer to our finding of weak cue-reward relationships (Line 21).

Signalling theory assumes that mutualists require a tight correlation between cue and reward (i.e., a high predictability)^{1,2}, in order to optimize their reward intake. Our study shows that even though fruit brightness explains only 11-35% of the variance in the nutrient composition of fruit pulp (after accounting for phylogenetic relatedness), birds rely on this information to select fruit species with a particular nutrient composition (as indicated by the structural equation model presented in Table 2 and Figure 2d). Note that regression models without phylogenetic information yielded similar results ($r^2 = 0.14$ — 0.29). The relationships reported in our study are weak when compared to a previous study about fruit choice of two warbler species in Mediterranean Scrubland, in which fruit colours explained 44-60% of variance in the nutrient composition of fruit pulp³. We added this comparison in the revised version of the discussion (Lines 227–230).

Related to the above, I feel that the non-expert reader simply needs to be provided a little bit more framework surrounding the import of this work. I learn from the text that this is the first time this has been shown, and that the issue is “highly contentious and unresolved” – but I do not get a good sense of what is at stake, thus making it difficult to assess the impact of this work for wider evolutionary biology. What are the implications of the findings here? Perhaps even more interesting, what would be the implications of weak associations being incapable

of leading to optimised intake – the hypothesis for which a counter example is now presented – e.g., what has been ruled out? How has this changed the understanding of the evolutionary trajectories of mutualistic networks? The authors need to more strongly state the case, and explain to the practitioners, of this work's interest to the wider field.

Response: Thank you for this critical comment. In response to your comment we expanded the first paragraph of the introduction to provide more background information (Lines 32–54) and also expanded the discussion to make the implications of our work more clear (Lines 218–226, Lines 227–259 and Lines 271–274).

In the revised version of the first paragraph (Lines 32–42), we now briefly characterize mutualisms (exchange of resources and services between species) and then outline that communication is assumed to structure interactions within mutualistic networks because the involved species often possess communicative traits that are adapted to stimulate the sensory system of their mutualistic partners. We provide pollination and seed dispersal mutualisms as specific examples. Then we characterize the different kinds of functions that communicative traits could fulfil, i.e., mere attraction of partners (by stimulation of their sensory system) and/or providing additional information to partners about the quality of rewards (via cues).

In the second paragraph (Lines 43–54), we now describe the controversy about the role of communication in mutualisms from the perspective of signalling theory: Theoretic models in signalling theory assume that partner choice requires highly reliable communication, but in many situations cues are characterized by low to moderate reliability. Despite the high prevalence and diversity of communicative traits in mutualisms, it is therefore unresolved to what extent communication contributes to partner choice and reward optimization in mutualistic networks.

In the revised version of the discussion, we now briefly discuss how the interplay of communication and nutrient regulation may contribute to explain the diversity of communicative traits in mutualisms (Lines 218–226). Moreover, we discuss which mechanisms might allow animal mutualists to optimize their rewards based on cues with low reliability and put our results in the context of other interaction systems such as mate choice and predator-prey interactions (Lines 227–259). We also mention the implications of our results in the context of biological market theory in the concluding paragraph (Lines 271–274). We hope that with these thorough revisions have adequately addressed your concerns.

Beyond the concern above, the paper presented a (mostly) clear and interesting read, and is an impressive use of multiple different data sets to inform theory in a novel way. The methodology appears rigorous and I am very happy (with my Bayesian hat on) with the statistical analyses. The data augmentation was described nicely and looks robust. I also very much appreciate the code for the analysis being provided in supplementary material.

Response: We appreciate these positive comments on the content of the manuscript and the statistical analyses.

Other major comments:

1. While the writing style is clear and straightforward, and the material mostly easily digestible, throughout the paper there is a pattern of presenting material slightly back-to-front, such that specific terms are used and then a few sentences later, their meaning is clarified (or occasionally only in the Methods/Supplementary Material). I would suggest the authors go through the paper with an eye to this, and make sure that any terms or references to things like methodology are actually understandable by the naïve reader at the moment of presentation. Please be aware that a reader of the journal is unlikely to flip to the methods or supplementary material each time one is mentioned: thus the text needs to be understandable at first read without the additional details found there. A few examples:

Response: Thank you for these suggestions. We went through the main manuscript and supplementary information and included missing information where necessary (see specific comments below).

a. p6 line 128: “the interaction strength of birds” is only defined in the Methods and has not been presented before this point even in vague terms as a feature of interest (perhaps because it turns out to have no association, but if it is tested, some rationale as to what it is and why it might be theoretically of interest should be provided).

Response: We added definitions of the three explanatory variables to the last paragraph of the introduction where we describe the main aims of the study (Lines 96–101). Moreover, to allow for a more intuitive interpretation of the term interaction strength, we adopted the terminology of Vazquez et al.⁴ and now use the term ‘frugivore impact (i.e., the relative

contribution of a frugivore to community-wide fruit removal)’. Frugivore impact quantifies the importance of a frugivore species in terms of its contribution to community-wide seed dispersal⁵. We now also outline in the revised introduction that the degree of generalization and the impact of frugivores are two key properties that do not only correlate with the effect of animals on network dynamics and plant regeneration processes^{6,7}, but also with the dependence of animals on plant resources^{8,9} (Lines 61–66). According to optimal foraging theory, generalist frugivores with a high impact in the networks should select high-caloric lipid-rich fruits to maximize their net energy gain during foraging, because these species depend strongly on fruit resources^{10,11}. We now state this hypothesis explicitly in the introduction (Lines 66–68).

b. p11-12 section “Frugivore traits”: this section follows the pattern of presenting undefined terms which are explained slightly later. Only subtle rewording would be required to fix such things, for example instead of “namely (i) the interaction strength, (ii) the proportional generalization (generalization hereafter)” I suggest something like “namely (i) the relative importance of a frugivore within the mutualism network, its interaction strength, (ii) a proportional measure of its dietary generalization (generalization hereafter)”. The description of both interaction strength and generalization brings the reader through a number of details before summarizing what the measure actually captures: better to start with this, so that the details make sense. It was only at the very end of the migratory distance part that I concluded that the study sites must be on breeding grounds; this should be clear earlier.

Response: We changed the order as suggested (Lines 362–368). The beginning of the section now reads: *“To test the hypothesis that the mean colour profile of consumed fruits and the mean nutrient intake of frugivorous birds are related to their generalization and impact in the networks and to their migratory behaviour we used three measures: (i) the diversity of plant species consumed by a frugivore species relative to the number of plant species available in a network (frugivore generalization), (ii) the relative contribution of a frugivore species to fruit removal in a network (frugivore impact), and (iii) the latitudinal distance between the study sites and the wintering range of a frugivore species (migratory behaviour).”* This introduction is then followed by the more technical descriptions.

c. p13 section “Modelling fruit colours...”: the word ‘Achromatic’ starts the sentence preceding the one that defines it: “There, we use the term ‘brightness’ when referring to

‘achromatic values’”; it ought to be defined first. Additionally, how “perception of fruit brightness” is measured is described right up in the front of this paragraph – I suggest introducing the term ‘achromatic’ here rather than after the discussion of it is over.

Response: We revised the section about fruit brightness. The section now begins with the description of the chromatic colour components (Lines 344–351) and is followed by a description of the achromatic component (Lines 351–354), which now reads “*We quantified perception of fruit brightness based on the excitation of the avian double cone using achromatic values (a). High achromatic values indicate that a fruit is perceived as being bright, whereas low values indicate that a fruit is perceived as being dark.*”

d. p14 line 14: referring to an equation several paragraphs before the equation leads to confusion – can the “Quantification” section be put before the Statistical analyses rather than interleaved into the submodels?

Response: As suggested we moved the “Quantification...” section right in front of the section about the “Statistical analyses” (Lines 388–403).

e. p17 and Fig 3b: it took quite a lot of thinking to figure out that the y-axis in figure 3b must represent 1-the proportion of colour profiles “inside the 95% confidence interval of the simulated colour profiles” (lines 411-412), because “non-random” must mean they are *outside* these confidence intervals. This whole analysis could be presented much more clearly.

Response: We are sorry for the lack of clarity in the description of the results of the Monte Carlo simulation in Figure 3b. In addition to your comment, reviewer #2 was concerned that only testing whether the brightness of consumed fruits differs from what would be expected by chance does not account for the fact birds might select fruits that are on average darker or brighter (with opposite nutritional consequences). Indeed, the only additional information provided by the Monte Carlo simulation was that non-random fruit choice of birds is more related to fruit brightness than to the chromatic colour components. We had originally included this simulation to support the results of the first and second objectives of the study (i.e., that nutrient content is mainly related to fruit brightness and that fruit choice of birds is mainly mediated by fruit brightness). However, as this third analysis is not essential for our

conclusions and was obviously rather causing confusion than improving the main statements we decided to remove this analysis in response the comment of the reviewer #2.

f. Supplementary Information p2: a selection of variables is presented on line 30 but not defined until line 38 – these could be moved closer (or inverted).

Response: Following your suggestion we slightly restructured the section so that variables are defined when they first occur in the text (Supplementary Methods 1, Lines 17–20).

Additional minor comments:

2. p5 line 95: suggest saying “An achromatic component” rather than “The achromatic component” since using the definite article suggests this is something or a component of something you’ve discussed already, yet it is an additional component beyond the tetrahedral shape defined by the chromatic components. (I spent a brief period of time trying to understand what particular chromatic colour component “The achromatic component” could possibly be, before continuing to read and having my wonder clarified: this is a very subtle form of the issue noted in comment 1, above.)

Response: We rephrased the sentence according to your suggestion (Lines 119–122).

3. p5 line 98 & p6 line 113: the fact that there is stochastic variable selection appears only in the supplementary material; a reader who looks through the methods for this phrase will be disappointed. I suggest referring to both Methods and Supplementary Methods here.

Response: We added a reference to the Supplementary Methods 1 in the results and methods section (Lines 125 & 414).

4. p11 line 254-255: I highly suspect that “the interaction frequency of each frugivore species” is meant to be something like “the number of interacting plants for each frugivore speices” – as frequency already means it’s divided by something! If not, where this frequency comes from needs to also be defined.

Response: We referred to the number of times a bird visited a particular plant species to feed on its fruits per hour (also see the comment of reviewer #3). We rephrased the sentence to make this more clear (Lines 287–289).

5. p15 equation (1): the variable J , the total number of frugivore species in the network, should be explicitly defined somewhere in the text.

Response: Reviewer #3 recognized that we mistakenly used the wrong subscripts in equations one and two (without effect on the model, where the equations were applied correctly). The sum operator should have been applied across the I plant species in a given network. We corrected the equations and added a definition for I (the total number of plant species in a network) to the text (Lines 389–403).

6. p16 lines 388-398: Could you please provide the outcome of these checks in the Supplementary Materials.

Response: We added two tables to the Supplementary Information, in which we provide the Potential Scale Reduction Factor (PSRF) as a measure of model convergence and the effective sample size (n_{eff}) for all main model parameters (see Supplementary Tables 6 and 7).

7. p17 line 413: should “differed between” be instead “differed among”? Unless the test was comparing the entirety of the chromatic components to brightness, but I don’t think this is the case. (And not pre-supposing your answer would be better there).

Response: As mentioned above, following the concerns of reviewer #2, we removed this analysis from the manuscript, because it is not essential for our main conclusions (see response to reviewer #2).

8. Tables and Figures: throughout the tables and figures, chromatic components are referred to in text as lower case letters $\{x,y,z\}$ but in the table/figure as upper case X, Y, Z (with the exception of the axes labels in Fig 2a,b). These should be made congruent in some manner – either with explanatory text (clunkier) or simply using upper or lowercase everywhere.

Response: We followed your suggestion and now refer to the chromatic and achromatic components of avian colour space (x, y, z and a) with lower case letters throughout the text, as well as in the tables and figures.

9. p26 Table 2 title: the term “phylogenetic path analysis” is not present anywhere else in the text, and while it can be figured out by reading in the Methods that the Bayesian hierarchical model is a path analysis and includes phylogeny as a random effect, at the time of reference to Table 2 the naïve reader has only seen a “Bayesian hierarchical model with a stochastic variable selection procedure” referred to: better to use a term more related to this.

Response: We changed the captions to “Bayesian hierarchical model” (Table 1) and “Bayesian hierarchical structural equation model” (Table 2) so that they are congruent with the text.

10. p29 line 648: Achromatic values {A} are referred to in reference to panels a and b, and yet they do not appear to be present in these panels? Either remove this reference or it might actually be possible to represent A as shades of grey in the plots?

Response: Thank you for this suggestion. We added shades of grey to the points in Figure 2a,b so that the achromatic values can be recognized.

11. p31 Fig 3a: Y and Z overlap in the presented plot. Can the labels to the dots be moved so they do not overlap?

Response: As suggested we moved the labels to avoid overlap.

12. Supplementary Information p3 description of equations 6-8: that I represents the identity matrix should be stated (assuming it does).

Response: Thank you for pointing to this lack of clarity. We added this information to the sentence (Supplementary Methods 1, Line 33).

13. Supplementary Information p8 Supplementary Table 1: Many undescribed abbreviations are presented; for example, the studies PL, DE, and GB should be re-identified here so this

can be referred to in isolation from the main text, and what do all the letters in the season column mean? (Oh, perhaps those are months?).

Response: We added the missing information to Supplementary Table 1.

14. Supplementary Information p8 Supplementary Table 1: I would much prefer if the actual data matrices tabulated could be provided in addition this summarizing information.

Response: We will make all data and code supporting the findings of our study available in figshare where we already reserved a DOI for the fileset (see Lines 493–501).

Reviewer #2 (Remarks to the Author):

General Comments

The authors address whether fruit brightness provides a reliable cue to birds about the nutritional quality of the fruits across several different ecosystems, and whether birds select fruits based on certain fruit traits. One of the more valuable components of this work (and a primary goal of this study) is the analysis across several plant-frugivore system of the relationship between fruit colour and nutrient composition of these fruits. Their novel conclusion is that the plant-frugivore mutualism can be maintained with only weak correspondence between the cue (fruit brightness) and the reward (nutritional quality). One of the less valuable components of this work (and the second goal of this study) is the attempt to relate fruit selection by frugivores to these same fruit characteristics. This latter goal falls short primarily because (a) the rationale provided is weak, and (b) the datasets used for evaluating this goal are incomplete and required too much estimation. These points are described below in more detail.

Response: We thank the reviewer for his/her very critical and constructive assessment of our manuscript, which in our opinion helped to improve the clarity and presentation of the work. Please see our responses to the detailed comments below.

The rationale for the work is quite broad (I would say too much so): mutualistic networks (they argue that the plant-frugivore interaction is mutualistic and thereby benefits the plant and frugivore) are maintained by reliable communication (in this case, fruit brightness communicates nutritional reward to the frugivore) between the partner mutualists. The conceptual problem with arguing that the system is a mutualistic network is in part related to the authors' comparison across the four seasons between resident birds (that they suggest select bright, lipid-rich fruits) and migratory birds (that they suggest select sugar- and antioxidant-rich fruits) for which the mutualistic relationship might be quite distinct. This suggests that the fruits selected by resident birds would be distinct from that of migrating birds which also means that the pair-wise mutualisms would also be distinct. However, the authors recognize that migratory birds might select different fruits across the four seasons, so they assume that migrants and residents have similar fruit preferences during the breeding season. How such seasonal changes in bird abundance, fruit phenology and abundance, and fruit selection results in true mutualisms is not well established in this manuscript. In sum, the reliance on a rationale related to mutualistic networks is weak and does not do justice to the interesting results.

Response: Obviously our rationale to use a network approach needs some clarification. First of all, our approach to analyse the pair-wise mutualistic relationships between plants and frugivores using network theory is nothing particularly new but builds on a solid theoretical framework⁶ that has been established during the last decades since the seminal work of Pedro Jordano (1987) in *American Naturalist*¹². This theoretical framework explicitly accounts for the fact that in the vast majority of cases mutualistic relationships are not exclusive one-to-one associations between pairs of species, but rather form heterogeneous networks of multispecific associations, in which most animal species interact with a range of plant species and share interaction partners with other animal species (and vice versa). In our opinion, this framework is perfectly suited to analyse community-wide patterns of fruit selection by frugivores in the context of communication and nutrient regulation.

Moreover, when reading the statement “*The conceptual problem with arguing that the system is a mutualistic network is in part related to the authors' comparison across the four seasons...*”, we had the impression that in the previous version of the manuscript, some aspects of our analysis were not described clear enough, which might have led to misunderstandings. It seems that the reviewer got the impression that we pooled all interaction data into one big network. This misunderstanding might stem from the fact that in

Figure 1 of the manuscript we graphically present a pooled network as a summary of the compiled interaction data (although in the figure caption we state that the analysis is based on individual networks). Yet, we are not treating the system as a single network in our analysis. On the contrary, our analysis explicitly builds on ten individual season- and site-specific plant-frugivore networks (we added this information to the caption of Figure 1 to avoid confusion). We do not use a pooled network to explicitly account for the fact that animals only have the opportunity to consume a particular plant species if they co-occur spatially and temporally (see statement in the Methods section, Lines 285–287).

Apart from this, a key advantage of the network approach that we have taken is that generalist and specialist frugivores, as well as migratory and resident frugivores co-occur within these seasonal networks and have access to the same fruiting plant species, because the data for each network have been recorded in the same place during the same time period (see the original publications for details). This setting represents a natural experiment that allows for a comparison of nutritional strategies between these groups, because interspecific differences in fruit selection of birds are not confounded by spatiotemporal constraints in resource availability. To make this more clear, we added the above information to the last paragraph of the introduction, where we introduce the study system (Lines 103–108). In this context, one of the most interesting aspects of our analysis is that given these equal opportunities, migratory and resident birds select fruits with a different nutrient composition.

It is true that we make the simplifying assumption that during the non-migration period (December to May) residents and migrants have similar colour preferences. (Note that those migrants that are present from December to May are mainly partial migrants, e.g. species of the genus *Turdus* that stay within Europe and shift their distribution southward.) However, in response to your comment, we also considered two alternative models. One model that included the generalization and impact of frugivores and their migratory distance, but without setting the migratory distance outside the pre-migration and migration periods to zero (alternative model 1); and a second model that included the generalization and impact of frugivores, their migratory distance, period (migration *versus* non-migration) and the interaction between migratory distance and period as fixed factors (alternative model 2). These two alternative models yielded identical conclusions regarding the fruit choice of resident and migratory birds (Table 2, Supplementary Tables 4 and 5). However, alternative model 1 had lower explanatory power ($r^2_m = 0.12$) than the model in which we set the migratory distance outside the pre-migration and migration periods to zero ($r^2_m = 0.17$; Table 2 and Supplementary Table 4). Moreover, in alternative model 2 neither period, nor its

interaction with migratory distance, were supported by the Bayesian indicator variable selection and the inclusion of both variables did not improve the explanatory power of the model ($r^2_m = 0.18$; Table 2 and Supplementary Table 5). Therefore, we report the results of the model in the main text, in which we set the migratory distance of migrants during the non-migration period to zero. In the section about statistical analyses, we now mention that we also fitted two alternative models that led to identical conclusions and report a summary of the model output in the supplementary materials (Supplementary Tables 4 and 5). We hope that our response to the reviewer's concerns as well as implementation in the manuscript text now clarifies our rationale and analyses.

We are not sure what the reviewer means by the sentence near the end of his/her comment: "*How such seasonal changes in bird abundance, fruit phenology and abundance, and fruit selection results in true mutualisms is not well established in this manuscript.*". We would appreciate if the reviewer could clarify the meaning of the sentence.

The three objectives outlined by the authors (lines 77-82) differ in their merit and are not consistently outlined throughout the manuscript. The first objective (assess the relationship between fruit colour and fruit nutrient composition) is well worth doing in part because this is done on a refreshingly broad scale (across 44 plant and 43 bird species). The second objective is confusing (not sure what this means: "preferences for particular nutrients are related to their generalization on fruit resources".) The third objective is trivial or too general – no vertebrate animal is expected to feed randomly, and testing whether birds "optimize their nutritional rewards" is not adequately informative (i.e., there is no discussion about the optimization criteria or how the fruit traits might translate into true 'rewards' for the frugivore – the latter requires knowing what nutrients are assimilated relative to the requirements of the birds). In fact, the 2nd and 3rd objectives are better described in the Stats Analyses section (lines 319-321): the 2nd hypothesis is that fruit choice by frugivores is mediated by fruit colour and related to frugivore traits. Although later in the methods this becomes hypotheses 2 and 3 (lines 366-367): fruit color mediates fruit choice and is related to frugivore traits. These latter statements of hypotheses (whether one or two) are a better characterization of what was done. In sum, the readers enthusiasm for the work waned by the end of the introduction because of the weak rationale, the heavy jargon (see Specific Comments below), and the lack of compelling objectives (except for the first one).

Response: We appreciate the interest of the reviewer in the first objective of our study and his/her constructive comments on the former objectives 2 and 3. Regarding the second objective, the wording of the sentence probably was unclear and we agree with the reviewer that this second goal is better described in the methods section. In response to your comment (and a comment of reviewer #1) we revised the part of the introduction, in which we summarize the main objectives to increase clarity (Lines 92–108). The objectives now read: *“Second, we test whether fruit choice of frugivorous birds is mediated by fruit colour and whether the frugivores’ mean intake of particular nutrients is related to their generalization and impact in the plant–frugivore networks (the diversity of plant species consumed by a frugivore and its relative contribution to fruit removal in a network, respectively) and to their migratory behaviour (the latitudinal migratory distance of a frugivore; see Methods for details).”* Thus, we also exchanged the term “frugivore traits” by the actual traits to be more precise. We hope that in doing so, we were able to justify the merit of our second objective. In fact, we are not aware of any study that has tried to link the fields of signalling theory (objective 1) with the fields of nutrient regulation and network theory (objective 2). Therefore, our second main finding that the nutrient intake of frugivores in species-rich mutualistic networks is related to certain biological characteristics of these species and in line with expectations from physiological experiments (see specific responses below) is not only novel, but also suggests that communication is an important mechanism of partner choice and reward optimization in mutualistic networks.

We agree with the comment of the reviewer regarding the third objective of the study (testing whether frugivores forage non-randomly). Indeed, the only additional information provided by the Monte Carlo simulation was that non-random fruit choice of birds is more related to fruit brightness than to the chromatic colour components. We had originally included this simulation to support the results of the first and second objectives of the study (i.e., that nutrient content is mainly related to fruit brightness and that fruit choice of birds is mainly mediated by fruit brightness). However, the reviewer is correct that only testing whether the brightness of consumed fruits differs from what would be expected by chance does not account for the fact birds might select fruits that are on average darker or brighter (with opposite nutritional consequences). As this third analysis is not essential for our conclusions and was obviously rather causing confusion than improving the main statements we decided to remove this analysis in response the comment of the reviewer.

The main results take advantage of previously compiled datasets on fruit characteristics, bird and plant phylogenies, bird vision capacity, and plant-frugivore visitation and consumption rates. The original results are the product of applying these previously compiled datasets to a meta-analysis of sorts that used a Bayesian hierarchical modeling framework. The completeness of these compiled datasets seemed adequate for testing hypothesis #1 (fruit colour related to nutrient composition) and found these results quite compelling. Unfortunately, the completeness of the datasets was not adequate for testing the other (or two) hypothesis (fruit choice related to fruit colour or nutrients). Most importantly, plant abundance data was only available for two of the ten networks (line 398) and so fruit consumption information was used to estimate plant abundance – this confounds availability and use information which is the basis for testing this 2nd (and 3rd) hypothesis. This undermines the reader's confidence in the network analyses on this point.

Response: As mentioned before, we appreciate the interest and confidence of the reviewer in the results of the first objective of the study. Yet, we disagree with the notion of the reviewer that the completeness of the dataset was not adequate to test the second hypothesis (*“fruit choice of frugivorous birds is mediated by fruit colour and the birds’ mean intake of particular nutrients is related to particular traits”*). Here again, a lack of clarity in the manuscript might have caused a misunderstanding about which kind of data was used to test the second hypothesis. The analyses for testing the second hypothesis did not require plant abundance data, as we only compared the mean colour values of consumed fruits and the mean nutrient intake of birds in relation to their generalization and impact in the networks and their migratory distance. As mentioned in a previous response, this comparison is possible because generalist and specialist frugivores, as well as migratory and resident frugivores co-occur within the seasonal networks and have access to the same fruiting plant species (because the data for each network have been recorded in the same place during the same time period). Therefore, even though we do not know about the relative abundance of the plant species in the networks, we know that given this natural abundance, generalist and specialist frugivores, as well as migratory and resident frugivores selected fruits of species with different brightness and nutrient content. These choices matched our expectations from previous physiological experiments (see detailed responses below). In addition, our conclusions are strengthened by the fact that the frugivore traits were solely related to fruit brightness, but not to the chromatic colour components of fruits. Therefore, the two (independent) analysis of (i) colour-reward relationships (Fig. 2c; Table 1) and (ii)

relationships between bird traits and colour profiles of consumed fruits (Fig. 2d; Table 2), together suggest that fruit brightness is an indicator of the nutrient content of fruits and that birds rely on this cue to regulate their nutrient intake (Fig. 3). Importantly, the data for frugivore traits and fruit colour were complete so that missing data does not compromise these main conclusions of our manuscript.

However, we absolutely agree with the reviewer that the test of the third hypothesis (*non-random foraging*) was compromised by the fact that we did not have plant abundance data for eight of the ten networks. As mentioned above, since this third analysis was not essential for our conclusions, but was only meant as an additional piece of information to support our main findings for objectives 1 and 2, we decided to remove this analysis from the manuscript. We think that removing hypothesis 3 added clarity to the text and appreciate the valuable comments.

Specific Comments

Lines 32-44: the jargon is quite heavy in this first paragraph to the point of confusion in several places. For example (#1), the statement "a fundamental question is whether communication among mutualists serves to attract partners via increased conspicuousness, or whether it also informs partner choice and reward optimization" does not adequately make clear to the general reader why these two parts (conspicuousness vs. reward optimization) are mutually exclusive, and what exactly is meant by "partner choice" and "reward optimization". For example (#2), this statement is quite dizzying and confusing even after reviewing the citation: "in most mutualisms individual fitness does not depend on the outcome of single erroneous interactions, but on the summed effect of repeated interactions" – needs better context.

Response: We agree with the reviewer that the general reader requires more background information here. Following your comment and the comment of reviewer #1 we thoroughly revised the first paragraph of the introduction to remove the heavy jargon and improve clarity. In the revised version of the first paragraph (Lines 32–42), we now briefly characterize mutualisms (exchange of resources and services between species) and then outline that communication is assumed to structure interactions within mutualistic networks because the species involved often possess communicative traits (e.g., colour and scent) that are adapted to stimulate the sensory system of their mutualistic partners. We provide pollination and seed

dispersal mutualisms as specific examples. Then we characterize the different kinds of functions that communicative traits could fulfil, i.e., mere attraction of partners (by stimulation of their sensory system) and/or providing additional information to partners about the quality of rewards (via cues). Thereby, the statement regarding the potential function (conspicuousness and/or reward optimization) in this paragraph does not imply that these two functions are mutually exclusive, as the second part of the sentence includes the conjunction ‘also’.

Regarding your second example: (“...*in most mutualisms individual fitness does not depend on the outcome of single erroneous interactions...*”). We moved this statement from the introduction to the discussion, where we have more space to discuss its implications for the contribution of weak cues to reward optimization in mutualisms (please see Lines 227–248 in the discussion). Instead in the revised version of the second paragraph of the introduction we now give a more general introduction to the controversy about communication in mutualisms (Lines 43–54).

Line 56: why would it be dietary preferences OR migratory behaviour? (birds during migration also exhibit strong diet preferences). The authors should also be careful about the word choice here: diet selection (use vs. availability) is distinct from diet preference (use given equal availability).

Response: Thank you for these suggestions. We removed the second part of the sentence (Lines 60–61). Moreover, we went through the text and exchanged the term ‘preference’ by ‘selection’ when we refer to our analysis of fruit choice by frugivorous birds.

Lines 59-61: this is an oversimplification. Birds (or most any other vertebrate) do not usually or commonly compensate for a protein deficit during migration or at other times of the annual cycle. The best examples come from the many, many studies of birds and mammals where energy density of diet is kept constant and the protein concentration is reduced (usually as carbohydrate is increased to keep the energy density constant) – these are the classic experiments where minimum protein requirements of an animal are defined. In such studies, animals usually eat the same amount of food on all diets (since they are isocaloric) but since the protein levels eventually are insufficient as dietary protein levels decline, there are signs of deficiency that allow the quantification of minimum protein requirements. Any animal

nutrition textbook outlines these more general patterns – my favorite for birds is Kirk Klasing's "Comparative Avian Nutrition" book (CAB International, UK; 1998).

Response: We appreciate this critical comment of the reviewer. We are aware of the comprehensive book by Kirk Klasing¹⁰. It is true that the book contains a chapter about the role of protein and information about the minimum protein requirements of a range of bird species (as determined by experiments; pages 140–150). However, to our knowledge the book does not contain an explicit statement about how low-protein diets (or a protein deficit) affect the food intake of birds. Our statement was based on experiments by Aamidor *et al.*¹³. In their study, Aamidor and colleagues experimentally tested how the availability of dietary protein in isocaloric diets influences food intake of migratory blackcaps (*Sylvia atricapilla*) during a migratory stopover. To do so, the authors did exactly what the reviewer suggested (reducing the protein concentration and increasing the concentration of carbohydrates while keeping energy density constant). Then the authors examined how the experimental manipulation of dietary protein content affected food intake and activity patterns. With this simple experiment the authors compellingly show that birds receiving a 3% protein diet increased their overall food intake compared to the group receiving a 20% protein diet. The authors concluded from their experiment that birds compensate for low dietary protein by behavioural responses (i.e., hyperphagia) that ensure rapid refuelling during migratory stopover. In response to this comment and your general concerns regarding our hypothesis for objective 2 (“*fruit choice of frugivorous birds is mediated by fruit colour and the birds’ mean intake of particular nutrients is related to particular traits*”), we thoroughly revised this paragraph to make the rationale behind our hypotheses more clear (see our next response).

Lines 66-73: these predictions about fruit traits (e.g., fat, carbohydrates, antioxidants) and their selection by resident vs. migrating birds is also oversimplified. Both groups of birds are likely maximizing their energy intake (not just resident birds as the authors argue), and all birds can fatten on fruits that are either high-carbohydrate or high-fat (birds that eat the former just need to use *de novo* lipogenesis to produce body fat). This undermines the primary rationale for comparing fruit selection in resident and migratory birds in this study.

Response: It is true that our explanation regarding the effect of lipid and carbohydrate on the accumulation of body fat in migratory birds was probably not clear enough. However, our hypothesis was based on previous physiological experiments by Smith & McWilliams¹⁴. In

their experiments, the authors simultaneously manipulated the protein, carbohydrate and lipid content of diets in a two-factorial design, while keeping the energy density fixed. Smith & McWilliams stress that only this experimental design allows disentangling the effects of all three macronutrients on fat metabolism and on the accumulation of body fat. Moreover, in the mentioned study the authors also monitored the effect of variation in the composition of dietary macronutrients on plasma lipid metabolite concentrations. In this regard the study provides a unique and comprehensive picture of the consequences of variation in the relative intake of macronutrients on the fat metabolism of songbirds that, to our knowledge, has not been achieved in previous experiments. In short, the authors show that, when dietary protein content is low (which is the case for fruit-based diets)¹⁰, birds on high-sugar diets gain more body fat than birds on high-lipid diets¹⁴. As the authors monitored plasma lipid metabolite concentrations of the birds, they were also able to show the underlying metabolic mechanism: The authors find that especially diets with high sugar content increase the rate and amount of fat deposition in migratory birds via hepatic *de novo lipogenesis*, whereas isoenergetic diets with high lipid content rather stimulate the direct utilization of dietary fat¹⁴. Therefore, birds on high-sugar, low-protein diets gain more body fat than birds on high-lipid, low-protein diets¹⁴. In light of these experiments, migratory birds should select sugar-rich fruits to efficiently gain body fat, whereas over-wintering, resident birds and birds that mainly rely on fruit resources (i.e., the generalist frugivores) should select lipid-rich fruits with an overall higher energy density to meet their metabolic demands.

Regarding the antioxidant hypothesis, experiments have shown that migratory birds are exposed to high oxidative stress associated with fat oxidation during flight¹⁵, and their innate immune function is compromised by physiological and energetic trade-offs¹⁶. We hypothesized that migratory birds should select anthocyanin-rich fruits, because previous experiments have shown that the intake of anthocyanin reduces oxidative stress and stimulates the immune response of birds^{15–20}.

To make our hypotheses clearer, we thoroughly revised this paragraph of the introduction (Lines 69–91).

Lines 145-147: this statement about the main novelty of this study is telling. I agree that the authors have shown weak associations between fruit colour and nutrient composition of fruits. I find little compelling evidence for the latter claim: frugivores "regulate their reward intake according to specific nutritional strategies". The next few sentences talk about "reward

optimization", "mutualistic networks", "partner choice", "functional adaptations" all of which are assumed and undocumented.

Response: Thank you. We have revised this first summarizing paragraph so that it is closer to our main findings (see Lines 177–186), and hope that by doing so we have adequately addressed the comment of the reviewer. The paragraph now reads: *“Our study provides the first community-wide assessment of the importance of cue–reward relationships for reward optimization in plant–animal mutualistic networks. We find weak associations of fruit nutritional rewards with fruit brightness, but no associations with chromatic components of fruit colouration. We further discover that, consistent with expectations from previous physiological experiments, the reward intake of frugivorous birds is related to their generalization in the networks and to their migratory behaviour and is primarily mediated by fruit brightness. These results suggest that frugivorous birds use the most reliable component of fruit colouration to discriminate the nutritional content of fruit pulp and that even weak cue–reward relationships may allow animal mutualists to optimize their reward intake according to specific nutritional strategies.”*

Reviewer #3 (Remarks to the Author):

Albrecht et al present a well-framed and written manuscript describing how weak cues from fruits drive frugivore selection and how this is related to the optimization of nutritional rewards. The authors compiled a large amount of high-quality data and they used the state of the art in statistical analyses to answer a very interesting question. In general, I greatly enjoyed the study and I think it may be of the interest of many other ecologists working in a wide range of fields. Also, except for some small questions on the methods (see below), the authors have included enough information to reproduce the study (given that they make the raw data available). I am describing further suggestions and questions related to specific lines:

Response: Thank you for this kind feedback on our manuscript.

Main text

- Title: The first time I read it I was not sure about what would be the content of the manuscript. I can totally understand that authors want to make the title and the study as much

generalizable as possible, but in this case the main issue of the study gets lost. I would suggest other more informative titles such as (but not limited to):

- o Reward optimization in mutualistic networks requires weak cue–reward relationships
- o Weak cues inform partner choice and reward optimization in mutualistic networks

Response: Thank you for these suggestions. We changed the title to ‘Reward optimization in mutualistic networks requires only weak cue–reward relationships’.

- Abstract, page 2, line 27 (P2L27): As written I thought you were going to show physiological experiments in the study. I’d rephrase to be clear that you are talking about previous studies. Maybe something like “consistent with previous physiological experiments”.

Response: We changed the sentence according to your suggestion (Line 27).

- P2L28-30: The authors write here and repeat several times in the text: “the extraordinary diversity of communicative traits in mutualistic networks originates from functional adaptations to inform partner choice and reward optimization”, also in P7L150-151 and in P10, L222-225: Even though this is a plausible explanation for the pattern, I don’t think their results demonstrate that. Several forces may be driving diversification in mutualist species communicative skills and their statement sound like an over-interpretation of their results. This may be suggested in the discussion as a possibility, but I would try to soften the affirmation and avoid including it in the abstract.

Response: Thank you for this constructive comment. We followed your suggestion and removed the statement from the abstract and rather highlight that given that animals require only weak cues to optimize their rewards, communication may be a common mechanism of partner choice and reward optimization in mutualistic networks (Lines 28–30). The last sentence of the abstract now reads: “*These results suggest that communication is a common mechanism of partner choice and reward optimization in mutualistic networks.*”

Moreover, in the discussion we refined and softened the argument regarding the consequences of functional adaptations to inform partner choice and reward optimization for the diversification of communicative traits (see Lines 218–226 & the next response below).

- P3L35-36: Related to the previous comment, why is this the only option? Why cannot this be at least partially caused by not other processes?

Response: We agree that this statement could have been more balanced. In the paragraph of the discussion where we mention the consequences of functional adaptations to reward optimization for the diversification of flower and fruit displays, we now also mention that other selective pressures (e.g., from antagonists) matter as well (Lines 224–226).

- P3L47: “The composition of fleshy fruits is often imbalanced...” what kind of imbalance are you referring to? In relation to what? Please be more specific, like it is written is ambiguous.

Response: We added some more information about which kind of nutrients we refer to (Lines 56–58). The sentence now reads: “*The macronutrient composition of fleshy fruits (i.e., the lipid, sugar and protein content of fruit pulp) is often unbalanced and ...*”. We hope this clarifies the meaning.

- P8L164-168: In this paragraph you say that the evolution of generalist mutualists may require both behavioral (...) adaptations and also “specific nutritional strategies that enable animal mutualists to rely solely on resources that they acquire through mutualistic interactions”. To be, this nutritional strategies are a kind of behavior. Maybe you should delete “behavioural” from line 166.

Response: We agree with your comment and rephrased the sentence (Line 204–208). The sentence now reads “*In addition, our results suggest that the morphological, physiological and behavioural adaptations of generalists in plant–animal mutualisms^{7,21–23} might also include specific nutritional strategies that enable them to rely mainly on resources that they acquire through mutualistic interactions (e.g., fruit pulp or nectar).*”.

- P9. In lines 189-192 you underline that cue-reward relationships can change geographically based on a study by Schaefer et al. (2014), and then in lines 207-210 you say that communication may be “remarkably robust to spatiotemporal variation”. This seems a bit contradictory to me. This made me think about whether the results were consistent among your 3 study areas. In the analyses you included study area as a random factor. I agree with

this, but this accounted for the between sites variability and we cannot really see how things changed among sites. I would like to see whether indeed results are similar for the different study areas. Maybe you can show the statistics for the random factor or do a simpler analysis where study area is included as a fixed factor just to see its effect. This will allow you to make more robust affirmations about the spatial variability.

Response: Thank you for this comment. Even though our statement that communication may be remarkably robust to spatiotemporal variation in cue-reward relationships seems a bit counter intuitive, previous work suggests that animals are indeed able to cope with spatiotemporal uncertainty in the information content of cues. In particular, previous experiments have shown that animals (vertebrates and insects) are able to adapt their foraging behaviour when the information content of cues is experimentally manipulated (e.g., by first allowing an animal to learn an association between a cue and a reward and then switching the ‘meaning’ of the cue associated with that reward)^{24,25}. This is possible, because animals can verify the accuracy and information content of cues after they have responded to them (e.g., after assessing the nectar volume of visited flowers or through post ingestive feedbacks and taste)^{26,27}. Therefore, the sensory and cognitive abilities of animals allow them to associate cues with rewards during repeated interactions, thus, allowing for highly adaptive foraging behaviour. This mechanism relaxes the assumption that reward optimization in mutualisms requires high (spatiotemporal) reliability. Therefore, we conclude that geographic variation in cue-reward relationships does not necessarily render weak cues useless for reward optimization by animals.

However, after reading your comment, we got the impression that this line of argument might rather confuse readers. Therefore, we would like to restructure our line of argument regarding the variation in cue-reward relationships in this paragraph. In particular, we think that it makes more sense to compare the strength rather than the direction of the relationships between our study and the study by Schaefer *et al.*³. This also satisfies the request of reviewer #1, who asked to put our conclusion that the colour-reward relationships across the three locations represent ‘weak cues’ in the context of previous work. Following this comparison, we discuss that patterns of fruit choice in terms of the brightness profile of consumed fruits varied only little between the three locations in our study (variance explained by site: $r^2_{site} = 0.10$; Supplementary Table 2). Then we discuss which mechanisms may allow birds to optimize their rewards despite the fact that cue-reward relationships are weak in our

study. In response to your comment, we now also provide a table with r^2 -values for all variance components associated with the random effects (Supplementary Table 2).

The revised paragraph of the discussion (Lines 227–248) now reads: “*In our study the strength of the relationships between fruit brightness and nutrients was rather low (variance explained by fruit colour after accounting for phylogenetic relatedness of plant species: $r^2_m = 0.11–0.35$) compared to a previous study about fruit choice of two warbler species in Mediterranean Scrubland ($r^2 = 0.44–0.60$)³. Nonetheless, patterns of fruit choice were highly consistent across the three localities in our study with little between-site variance in the brightness profile of selected fruits (variance explained by site: $r^2_{site} = 0.10$; Supplementary Table 2). This suggests that birds are able to optimize their rewards despite high uncertainty in colour–reward relationships. Because visual discrimination is only the first step of decision-making during foraging²⁸, animal mutualists may respond to uncertainty in cue–reward relationships by relying on taste, post-ingestive feedbacks or other mechanisms to verify the reliability of cue–reward relationships and adapt their foraging behaviour accordingly^{24–27,29,30}. Supporting this idea, previous studies found that flower visiting insects adapt their foraging behaviour in response to intra- and inter-individual variation in cue–reward relationships by ceasing interactions with plants whose cues are inaccurate or misleading^{25,27}. This may also pose a mechanism for selection on reliability of cue–reward relationships^{27,30,31}. The verification of interaction outcomes is possible, because mutualisms are typically characterized by repeated interactions among partners. In this situation individual fitness usually does not depend on the outcome of a single interaction, but on the cumulative effects of repeated interactions^{3,5}. Therefore, the self-serving behaviour of animal mutualists may result in fair trade, as long as the cumulative effects of repeated highly beneficial interactions on individual fitness offset the negative fitness consequences of a few less beneficial interactions^{29,32}.”*

We hope that with these revisions we adequately addressed the concerns of the reviewer.

Results

- P11L241: How were visitation rates calculated? Number of visits/hour? Please be specific, it is easy to get lost with so many calculations.

Response: The reviewer is correct. We calculated visitation rates as the number of visits per hour. We added this information as well as information for fruit consumption rate (number of fruits removed per hour) in the respective paragraph (Lines 287–302).

- P11L241-244: If, as you say in the last sentence of P10 and the first one of P11, you only compiled information from DE and PL about the number of fruits consumed/visit, how did you calculate the rate of fruit removal for the GB data?

Response: Thank you for drawing our attention to this lack of clarity. Stiebel & Bairlein³³ (DE hereafter) and Albrecht *et al.*³⁴ (PL hereafter) recorded data on the number of fruits consumed per visit for each plant–frugivore pair (fruit consumption rate per visit hereafter)³⁵. This data included information for 34 of the 44 plant species and for 39 of the 43 frugivore species in the dataset. For plant–frugivore pairs in GB (Snow & Snow³⁶) for which we had data on the fruit consumption rate per visit from DE and PL we used the data from these two localities. When no information about the fruit consumption rate per visit was available for a plant–frugivore pair, we used the mean fruit consumption rate per visit of the frugivore species on other plant species. For four frugivore species without any information on the fruit consumption rate per visit we used the mean value across the remaining 39 frugivore species. We multiplied the visitation rate per hour with the fruit consumption rate per visit to estimate the fruit consumption rate of each frugivore species on each plant species in each network per hour (fruit consumption rate per hour hereafter; GB: $n = 111,434$ estimated total number of fruits removed; DE: $n = 31,789$; PL: $n = 16,366$).

To assess the sensitivity of our results to this approach, we conducted the analysis twice, once using networks based on fruit consumption rates per hour and once using networks based on visitation rates per hour. The results of both analyses were virtually identical (Table 2, Supplementary Table 3). This suggests that our conclusions are not affected by the uncertainty associated with missing information on the fruit consumption rate per visit for one of the three localities or by the decision of whether to use fruit consumption or visitation rates per hour as the ‘interaction currency’. This is due to the fact that the comparatively large variation in the visitation rates of a frugivore species across different plant species overrides the comparatively small variation in the fruit consumption rate per visit on each plant species⁵. Therefore, the visitation rate of frugivores on plants per hour is a strong predictor of their fruit consumption rates on these plants per hour, regardless of the

number of fruits consumed per visit (Supplementary Fig. 1). In the main text we report results based on fruit consumption rates per hour.

We added this information and the results of the sensitivity analysis to the methods section and to the supplementary materials (Lines 278–315, Supplementary Table 3 and Supplementary Fig. 1).

- P15 equations 1 & 2: I think that there may be either something missing in the equation or in the description. Equation 1 & 2 seem to give the color and nutritional profile of a specific fruit i , and not the “mean color/nutritional profile of the fruits that bird j consumed”. I think the equation needs to be completed to express the mean of all the plants.

Response: Thank you. In both equations, the sum operator was indexed in a wrong way (j in 1 to J instead of i in 1 to I) and the sum operator in the numerator was missing. We corrected this error in the formulas (Lines 395 and 401). In the model, we applied the correct formulas, so that our results are not affected.

- P15L363: I guess the “pulp dry mass of plant species” is the pulp dry mass per fruit of each plant species?

Response: This is correct. We changed the wording (Line 402).

- P17L409. I think the n_j should be c_j here.

Response: We are sorry for the lack of clarity in the description of the Monte Carlo simulation in Methods section and in Figure 3b. In addition to your comments, reviewer #2 was concerned that only testing whether the brightness of consumed fruits differs from what would be expected by chance does not account for the fact birds might select fruits that are on average darker or brighter (with opposite nutritional consequences). Indeed, the only additional information provided by the Monte Carlo simulation was that non-random fruit choice of birds is more related to fruit brightness than to the chromatic colour components. We had originally included this simulation to support the results of the first and second objectives of the study (i.e., that nutrient content is mainly related to fruit brightness and that fruit choice of birds is mainly mediated by fruit brightness). However, as this third analysis is not essential for our conclusions and was obviously rather causing confusion than improving

the main statements we decided to remove this analysis in response the comment of the reviewer #2.

- Page 17: In your randomization approach, I am wondering how good was the estimation of the relative availability of the plant species. Maybe you can compare the results from the simulation with the data for the two study sites for which you had abundance data. Are they highly correlated? Alternatively, you can repeat the analysis using only the two sites for which you had data on abundance and see if the results are consistent.

Response: We followed your suggestion and assessed how strongly fruit removal is correlated with the abundance of the plants in the two networks. Fruit removal was highly correlated with the relative abundance (crop size) of the plant species in the two networks for which we had data for both variables (Spearman's rank correlation $r_s = 0.68$, $P < 0.05$, $n = 13$ plant species and $r_s = 0.95$, $P < 0.01$, $n = 8$), indicating that the total number of fruits removed per hour is a suitable proxy for the relative abundance of the plant species in the networks. However, as mentioned above, in response to the concerns of reviewer #2, we decided to remove this analysis from the manuscript, because it was not essential for our main conclusions (see responses to reviewer #2).

- P17L412: I am not sure what do you mean for "(random: true versus false)" and (interaction term: random x color component), can you please explain it a bit more?

Response: As mentioned above, following the concerns of reviewer #2, we removed this analysis from the manuscript, because it was not essential for our main conclusions (see response to reviewer #2).

- Figure 1. Very nice and informative figure! I'd also describe in the caption or in the figure what x,y,z and A are so that reader do not need to go to the text, and also what GB, DE and PL stand for.

Response: We added information about the variables x, y, z and a to the caption of Figure 1. In addition, we slightly rephrased the reference to the abbreviations for the three study locations in the figure caption to improve clarity.

- Figure 3b: I think this figure needs to be explained a bit better, I am still not 100% sure of what it is representing, even though the results described in lines 138-142 are clear.

Response: In response to the concerns of reviewer #2, we removed this analysis from the manuscript, because it was not essential for our main conclusions (see response to reviewer #2).

Cited references

1. Edwards, D. P. & Yu, D. W. The roles of sensory traps in the origin, maintenance, and breakdown of mutualism. *Behavioral Ecology and Sociobiology* **61**, 1321–1327 (2007).
2. Hammerstein, P. & Noë, R. Biological trade and markets. *Philosophical Transactions of the Royal Society B: Biological Sciences* **371**, 20150101 (2016).
3. Schaefer, H. M., Valido, A. & Jordano, P. Birds see the true colours of fruits to live off the fat of the land. *Proceedings of the Royal Society B: Biological Sciences* **281**, 20132516–20132516 (2014).
4. Vázquez, D. P. *et al.* The strength of plant-pollinator interactions. *Ecology* **93**, 719–725 (2012).
5. Vázquez, D. P., Morris, W. F. & Jordano, P. Interaction frequency as a surrogate for the total effect of animal mutualists on plants. *Ecology Letters* **8**, 1088–1094 (2005).
6. Bascompte, J. & Jordano, P. *Mutualistic networks*. (Princeton University Press, 2014).
7. Guimarães, P. R., Jordano, P. & Thompson, J. N. Evolution and coevolution in mutualistic networks. *Ecology Letters* **14**, 877–885 (2011).
8. Fricke, E. C., Tewksbury, J. J., Wandrag, E. M. & Rogers, H. S. Mutualistic strategies minimize coextinction in plant–disperser networks. *Proceedings of the Royal Society B: Biological Sciences* **284**, 20162302 (2017).
9. Dalsgaard, B. *et al.* Opposed latitudinal patterns of network-derived and dietary specialization in avian plant-frugivore interaction systems. *Ecography* 1–7 (2016). doi:10.1111/ecog.02604
10. Klasing, K. C. *Comparative avian nutrition*. (Cab International, 1998).
11. Pyke, G. H., Pulliam, H. R. & Charnov, E. L. Optimal foraging: A selective review of theory and tests. *The Quarterly Review of Biology* **52**, 137–154 (1977).
12. Jordano, P. Patterns of mutualistic interactions in pollination and seed dispersal: connectance, dependence asymmetries, and coevolution. *The American Naturalist* **129**, 657–677 (1987).

13. Aamidor, S. E., Bauchinger, U., Mizrahy, O., McWilliams, S. R. & Pinshow, B. During stopover, migrating blackcaps adjust behavior and intake of food depending on the content of protein in their diets. *Integrative and Comparative Biology* **51**, 385–393 (2011).
14. Smith, S. B. & McWilliams, S. R. Dietary macronutrients affect lipid metabolites and body composition of a migratory passerine, the White-throated sparrow (*Zonotrichia albicollis*). *Physiological and Biochemical Zoology* **82**, 258–269 (2009).
15. Skrip, M. M. *et al.* Migrating songbirds on stopover prepare for, and recover from, oxidative challenges posed by long-distance flight. *Ecology and Evolution* **5**, 3198–3209 (2015).
16. Eikenaar, C. & Hegemann, A. Migratory common blackbirds have lower innate immune function during autumn migration than resident conspecifics. *Biology Letters* **12**, 78–81 (2016).
17. Catoni, C., Schaefer, H. M. & Peters, A. Fruit for health: the effect of flavonoids on humoral immune response and food selection in a frugivorous bird. *Functional Ecology* **22**, 649–654 (2008).
18. Bolser, J. A. *et al.* Birds select fruits with more anthocyanins and phenolic compounds during autumn migration. *The Wilson Journal of Ornithology* **125**, 97–108 (2013).
19. Cooper-Mullin, C. & McWilliams, S. R. The role of the antioxidant system during intense endurance exercise: lessons from migrating birds. *The Journal of Experimental Biology* **219**, 3684–3695 (2016).
20. Schaefer, H., McGraw, K. & Catoni, C. Birds use fruit colour as honest signal of dietary antioxidant rewards. *Functional Ecology* **22**, 303–310 (2008).
21. Carnicer, J., Jordano, P. & Melian, C. J. The temporal dynamics of resource use by frugivorous birds: a network approach. *Ecology* **90**, 1958–1970 (2009).
22. Jordano, P. Frugivory, external morphology and digestive system in mediterranean sylviid warblers *Sylvia* spp. *Ibis* **129**, 175–189 (1987).
23. Carnicer, J., Abrams, P. A. & Jordano, P. Switching behavior, coexistence and diversification: comparing empirical community-wide evidence with theoretical predictions. *Ecology Letters* **11**, 802–808 (2008).
24. Schaefer, H. M., Spitzer, K. & Bairlein, F. Long-term effects of previous experience determine nutrient discrimination abilities in birds. *Frontiers in Zoology* **5**, 4 (2008).
25. Real, L. Animal choice behavior and the evolution of cognitive architecture. *Science* **253**, 980–986 (1991).

26. Schaefer, H. M., Schmidt, V. & Bairlein, F. Discrimination abilities for nutrients: which difference matters for choosy birds and why? *Animal Behaviour* **65**, 531–541 (2003).
27. Brandenburg, A., Kuhlemeier, C. & Bshary, R. Hawkmoth pollinators decrease seed set of a low-nectar *Petunia axillaris* line through reduced probing time. *Current Biology* **22**, 1635–1639 (2012).
28. Sallabanks, R. Hierarchical mechanisms of fruit selection by an avian frugivore. *Ecology* **74**, 1326–1336 (1993).
29. Broom, M., Ruxton, G. D. & Schaefer, H. M. Signal verification can promote reliable signalling. *Proceedings of the Royal Society B: Biological Sciences* **280**, 20131560–20131560 (2013).
30. Benitez-Vieyra, S., Ordano, M., Fornoni, J., Boege, K. & Domínguez, C. A. Selection on signal-reward correlation: limits and opportunities to the evolution of deceit in *Turnera ulmifolia* L. *Journal of Evolutionary Biology* **23**, 2760–2767 (2010).
31. Benitez-Vieyra, S., Fornoni, J., Perez-Alquicira, J., Boege, K. & Dominguez, C. A. The evolution of signal-reward correlations in bee- and hummingbird-pollinated species of *Salvia*. *Proceedings of the Royal Society B: Biological Sciences* **281**, 20132934–20132934 (2014).
32. Jones, E. I. *et al.* Cheaters must prosper: reconciling theoretical and empirical perspectives on cheating in mutualism. *Ecology Letters* **18**, 1270–1284 (2015).
33. Stiebel, H. & Bairlein, F. Frugivorie mitteleuropäischer Vögel I: Nahrung und Nahrungserwerb. *Vogelwarte* **46**, 1–23 (2008).
34. Albrecht, J. *et al.* Variation in neighbourhood context shapes frugivore-mediated facilitation and competition among co-dispersed plant species. *Journal of Ecology* **103**, 526–536 (2015).
35. Schupp, E. W., Jordano, P. & Gómez, J. M. A general framework for effectiveness concepts in mutualisms. *Ecology Letters* **20**, 577–590 (2017).
36. Snow, B. K. & Snow, D. *Birds and berries: a study of an ecological interaction*. (T & A D Poyser, 1988).

Reviewers' Comments:

Reviewer #1:

Remarks to the Author:

This is a revision of a paper I've seen before. The authors have done an excellent job of addressing my original main issue, understanding what a "weak" interaction means, both in the introduction and further in the discussion. The introduction has also been expanded to provide a greater level of background to the non-expert reader, and I feel I can much better understand the place of this work in the literature and its import.

One part I feel could still be expanded is in the comparison to the previous study with a tighter (stronger) relationship: the current study shows that reward optimisation can happen even with weak relationships, but is there any idea of why the other study found something stronger? What features might lead to less or more tightness in cue-reward signals?

I very much like the discussion about 'one-time interaction' and 'repeated interactions'. The first two hypotheses appear quite logical. However, I'm not entirely convinced that more cheating is a clear outcome of the first two hypotheses - unless cheating and gaining a 'first' benefit from naive partners is enough to offset the later avoidance by experienced partners. Thus, I'm not sure a blanket statement about it being more common in 'repeated interactions' systems is fully supportable; it is more likely a balance between the needs and benefits of each side - for whom does a single interaction have greater payoff? I am not sure how easy this might be to present in a single sentence, and the statement addresses that modelling will likely be needed to tease out details in any case. But it may be worth considering how strongly the authors feel this is a clear prediction, or whether they wish to qualify it some.

I also appreciate the authors explicitly addressing the fields they are combining (network theory, signalling theory, and nutritional ecology), which makes more clear how they have been able to come at the question from the novel direction they do.

Overall, the writing is much clearer and many small confusions and typos have been corrected. I feel I can read through the text and understand the study much better now (and the removal of the randomisation analysis also assists here). I remain happy with the thoroughness of the statistical analysis and appreciate the additional details presented in the Supplementary Materials.

One final comment would be with the revised final statement in the abstract - while this is a softening of a previous statement, it does still seem to go a bit beyond the results by stating that communication is a common mechanism. It is hard to generalise that something is "common" from a single example, even if this includes a network of a large number of species. The final sentence in the introduction (and the text of the rebuttal) use the verb "may be" which could potentially soften it just enough more to be believable.

Reviewer #3:

Remarks to the Author:

I have read the review of the paper "Reward optimization in mutualistic networks requires only weak cue-reward relationships" by Dr Albrecht et al. and I think the manuscript has improved significantly from the previous version. I have only one further comment. I am not very convinced with the terms "generalization" and "impact" used. Basically, I think they are not very self-explanatory and I had to turn back to the definitions a couple of times during the reading to remember what they were

measuring. For "generalization", I'd use something like "diversity of interacting partners", while for impact, I like better the term used in the previous version "interaction strength", or even "contribution to fruit removal". I know these are longer, but I think this will increase the flow of the reading. Congrats for the nice work!

We thank the Reviewers for their constructive comments. Below we provide a point-by-point response to the comments, which are highlighted in red font. To facilitate the review process, we have highlighted the changes made to the manuscript with track changes in the Word document. Please note that line numbers in our responses refer to lines in the revised version of the manuscript with track changes switched on.

REVIEWERS' COMMENTS:

Reviewer #1 (Remarks to the Author):

This is a revision of a paper I've seen before. The authors have done an excellent job of addressing my original main issue, understanding what a "weak" interaction means, both in the introduction and further in the discussion. The introduction has also been expanded to provide a greater level of background to the non-expert reader, and I feel I can much better understand the place of this work in the literature and its import.

Response: We are happy that the revised version of the manuscript satisfies the reviewer.

One part I feel could still be expanded is in the comparison to the previous study with a tighter (stronger) relationship: the current study shows that reward optimisation can happen even with weak relationships, but is there any idea of why the other study found something stronger? What features might lead to less or more tightness is in cue-reward signals?

Response: We added potential explanations as to why the strength of the colour-reward relationships may differ between studies and a potential avenue for future research (Lines 298-304).

I very much like the discussion about 'one-time interaction' and 'repeated interactions'. The first two hypotheses appear quite logical. However, I'm not entirely convinced that more cheating is a clear outcome of the first two hypotheses - unless cheating and gaining a 'first' benefit from naive partners is enough to offset the later avoidance by experienced partners. Thus, I'm not sure a blanket statement about it being more common in 'repeated interactions' systems is fully supportable; it is more likely a balance between the needs and benefits of each side - for whom does a single interaction have greater payoff? I am not sure how easy

this might be to present in a single sentence, and the statement addresses that modelling will likely be needed to tease out details in any case. But it may be worth considering how strongly the authors feel this is a clear prediction, or whether they wish to qualify it some.

Response: Thank you for this comment. In response to your suggestion and the concerns of the editor, we revised the respective paragraph so that the perspective is more balanced (Lines 319-345).

I also appreciate the authors explicitly addressing the fields they are combining (network theory, signalling theory, and nutritional ecology), which makes more clear how they have been able to come at the question from the novel direction they do.

Overall, the writing is much clearer and many small confusions and typos have been corrected. I feel I can read through the text and understand the study much better now (and the removal of the randomisation analysis also assists here). I remain happy with the thoroughness of the statistical analysis and appreciate the additional details presented in the Supplementary Materials.

Response: We appreciate these positive comments.

One final comment would be with the revised final statement in the abstract - while this is a softening of a previous statement, it does still seem to go a bit beyond the results by stating that communication is a common mechanism. It is hard to generalise that something is "common" from a single example, even if this includes a network of a large number of species. The final sentence in the introduction (and the text of the rebuttal) use the verb "may be" which could potentially soften it just enough more to be believable.

Response: In response to your concerns and the request of the editor, we softened our main conclusion in the abstract, introduction and discussion (Lines 31-33; Lines 152-154; Lines 244-247; Lines 381-384)

Reviewer #3 (Remarks to the Author):

I have read the review of the paper “Reward optimization in mutualistic networks requires only weak cue–reward relationships” by Dr Albrech et al. and I think the manuscript has improved significantly from the previous version. I have only one further comment. I am not very convinced with the terms “generalization” and “impact” used. Basically, I think they are not very self-explanatory and I had to turn back to the definitions a couple of times during the reading to remember what they were measuring. For “generalization”, I’d use something like “diversity of interacting partners”, while for impact, I like better the term used in the previous version “interaction strength”, or even “contribution to fruit removal”. I know these are longer, but I think this will increase the flow of the reading.

Congrats for the nice work!

Response: We appreciate the positive comments by the reviewer. We followed the reviewer’s suggestion and changed the terms “generalization” and “impact” to “partner diversity” and “interaction strength” throughout the manuscript.